# FastFill: Efficient Compatible Model Update

**Florian Jaeckle**[*]
University of Oxford[†]

**Fartash Faghri**
Apple

**Ali Farhadi**
Apple

**Oncel Tuzel**
Apple

**Hadi Pouransari**[*]
Apple

## Abstract

In many retrieval systems the original high dimensional data (e.g., images) is mapped to a lower dimensional feature through a learned embedding model. The task of retrieving the most similar data from a gallery set to a given query data is performed through a similarity comparison on features. When the embedding model is updated, it might produce features that are not comparable/compatible with features already in the gallery computed with the old model. Subsequently, all features in the gallery need to be re-computed using the new embedding model – a computationally expensive process called *backfilling*. Recently, compatible representation learning methods have been proposed to avoid backfilling. Despite their relative success, there is an inherent trade-off between the new model performance and its compatibility with the old model. In this work, we introduce FastFill: a compatible model update process using feature alignment and policy based partial backfilling to promptly elevate retrieval performance. We show that previous backfilling strategies suffer from decreased performance and demonstrate the importance of both the training objective and the ordering in *online* partial backfilling. We propose a new training method for feature alignment between old and new embedding models using uncertainty estimation. Compared to previous works, we obtain significantly improved backfilling results on a variety of datasets: mAP on ImageNet (+4.4%), Places-365 (+2.7%), and VGG-Face2 (+1.3%). Further, we demonstrate that when updating a biased model with FastFill, the minority subgroup accuracy gap promptly vanishes with a small fraction of partial backfilling.[1]

## 1 Introduction

Retrieval problems have become increasingly popular for many real-life applications such as face recognition, voice recognition, image localization, and object identification. In an image retrieval setup, we have a large set of images called the gallery set with predicted labels and a set of unknown query images. The aim of image retrieval is to match query images to related images in the gallery set, ideally of the same class/identity. In practice, we use low-dimensional feature vectors generated by a learned embedding model instead of the original high dimensional images to perform retrieval.

When we get access to more or better training data, model architectures, or training regimes we want to update the embedding model to improve the performance of the downstream retrieval task. However, different neural networks rarely learn to generate compatible features even when they have been trained on the same dataset, with the same optimization method, and have the same architecture (Li et al., 2015). Hence, computing the query features with a new embedding model, whilst keeping the old gallery features, leads to poor retrieval results due to incompatibility of old and new embedding models (Shen et al., 2020).

---

[*]Corr: florian.jaeckle@gmail.com & mpouransari@apple.com

[†]Work completed during internship at Apple.

[1]Code available at https://github.com/apple/ml-fct.

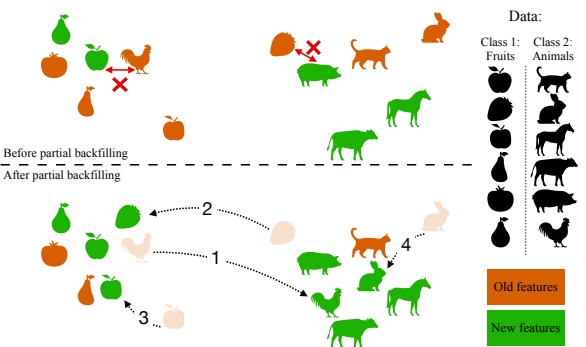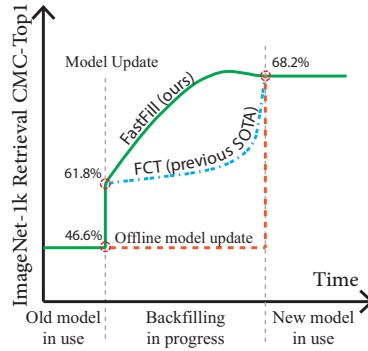

Figure 1: **Left:** Old and new features for a binary fruits vs. animals classification setup. Old and new features are somewhat compatible, but have a few mismatches shown by red crosses (top). We can improve retrieval performance by backfilling some of the old features to higher quality new features in a specific order (bottom). In a partial backfilling scenario the accuracy improvement depends on the order of backfilling. **Right:** Retrieval performance for ImageNet as a function of time when updating the embedding model with different backfilling strategies. A backfilling curve reaching the new model performance faster is better.

The vanilla solution is to replace the features in the gallery set that have been generated by the old model with features from the new model. This computationally expensive process is called *backfilling*. In practice, we carry out backfilling *offline*: we carry on using the old gallery features and the old model for queries whilst computing new gallery features with the new model in the background (see Figure 1-right). We only use the new gallery features once we've finished updating the entire set. However, for large scale systems this process is computationally expensive and can take months. In many real world systems, the cost of backfilling is a blocker for model updates, despite the availability of a more accurate model.

This has led rise to the study of compatible representation learning. Shen et al. (2020); Budnik & Avrithis (2021); Zhang et al. (2021); Ramanujan et al. (2022); Hu et al. (2022); Zhao et al. (2022); Duggal et al. (2021) all proposed methods to update the embedding model to achieve better performance whilst still being compatible with features generated by the old model (see Figure 1-left-top). Despite relative success, compatibility learning is not perfect: performing retrieval with a mixture of old and new features achieves lower accuracies than when we replace all the old features with new ones. In this work, we focus on closing this performance gap. Further, some previous methods degrade the new model performance when trying to make it more compatible with the old model (Shen et al., 2020; Hu et al., 2022) or requiring availability of side-information, extra features from a separate self-supervised model, (Ramanujan et al., 2022) (which may not be available for an existing system). We relax both constraints in this work.

To benefit from the more accurate embedding model sooner and cheaper, we can backfill some or all of the images in an *online* continuous fashion: We run downstream retrieval tasks on a partially backfilled gallery set where part of the features are computed with the old model, and part of them with the new model (see Figure 1-left-bottom).

In practice, we can consider two scenarios: 1) we will backfill the entire gallery set with a *random order* in an extended period of time and want to maximize the average performance during the backfilling process (see Figure 1-right); 2) we have a fixed partial backfilling budget and want to reach optimal retrieval performance after backfilling the allowed number of images (e.g., we can backfill only 10% of the gallery). In both cases, we want to reach the highest possible performance by backfilling the fewest images possible. We demonstrate that the training objective as well as the order by which we backfill images are both crucial. In fact, if we use training losses proposed in the literature and choose a random ordering we may even reduce performance before we see an improvement (see FCT-Random in Figure 2a) due to the occurrence of 'negative flips' (Zhang et al., 2021) - images that where classified correctly when using the old model but are misclassified when using the new one.

In this work, we propose FastFill: an efficient compatible representation learning and online backfilling method resulting in prompt retrieval accuracy boosts as shown in Figure 1-right. In FastFill, we learn a computationally efficient holistic transformation to align feature vectors from the old model embedding space to that of new model. The new model is trained independently without sacrificing its performance for compatibility. We propose a new training loss that contains point-to-point and point-to-cluster objectives. We train the transformation model with the proposed loss using uncertainty estimation and obtain significantly improved partial backfilling performance compared to previous works.

We summarize our main contribution as below:

- We propose FastFill, a new compatible training method with improved retrieval performance at all levels of partial backfilling compared to previous works.
- FastFill uses a novel training objective and a new policy to order samples to be backfilled which we show to be critical for improved performance.
- We demonstrate the application of FastFill to update biased embedding models with minimum compute, crucial to efficiently fix large-scale biased retrieval systems.
- On a variety of datasets we demonstrate FastFill obtains state-of-the-art results: mAP on ImageNet (+4.4%), Places-365 (+2.7%), and VGG-Face2 (+1.3%).

## 2 PROBLEM SETUP

**Image Retrieval** In an image retrieval system, we have a set of images called the gallery set $\mathcal{G}$ with predicted labels, which can be separated into $k$ different classes or clusters. At inference time, we further have a query set $\mathcal{Q}$. For each image in the query set we aim to retrieve an image from the gallery set of the same class. In an embedding based retrieval system, we replace the original $D$-dimensional data ($3 \times h \times w$ for images) in the gallery and query sets with features generated by a trained embedding model $\phi : \mathbb{R}^D \mapsto \mathbb{R}^d$, where $d \ll D$. The model $\phi$ is trained on images $\boldsymbol{x} \in \mathcal{D}$, a disjoint dataset from both $\mathcal{G}$ and $\mathcal{Q}$. For a query image $\boldsymbol{x}_q \in \mathcal{Q}$, the retrieval algorithm returns image $\boldsymbol{x}_i$ from the gallery satisfying:

$$\text{(single-model retrieval)} \quad \boldsymbol{x}_i = \arg\min_{\boldsymbol{x}_j \in \mathcal{G}} \mathfrak{D}(\phi(\boldsymbol{x}_j), \phi(\boldsymbol{x}_q)), \tag{1}$$

for some distance function $\mathfrak{D}$ (e.g., $l_2$ or cosine distance). In this setup we only need to compute the features $\phi(\boldsymbol{x})$ of the gallery set once and store them for future queries. We then no longer need to be able to access the original images of the gallery. As $d \ll D$, replacing the images with their features greatly reduces the memory and computational requirement to maintain the gallery and perform retrieval, crucial for private on-device retrieval. Depending on the context, by gallery we may refer to the original images or the corresponding features.

**Learning Compatible Representations.** For real world applications, we aim to update the embedding model every few months to improve its accuracy 1) to adapt to changes in the world, e.g., supporting people with face mask for a face recognition system, 2) on under-represented subgroups, e.g., new ethnicity, new lighting condition, different data capture sensors, by adding more data to the training set, or 3) by using enhanced architecture and training optimization. We denote the old and new embedding models by $\phi_{old}$ and $\phi_{new}$ and the training datasets of the two models by $\mathcal{D}_{old}$ and $\mathcal{D}_{new}$, respectively. After a model update, for every new query image we use $\phi_{new}$ to compute the feature, and hence the retrieval task turns in to:

$$\text{(cross-model retrieval)} \quad \boldsymbol{x}_i = \arg\min_{\boldsymbol{x}_j \in \mathcal{G}} \mathfrak{D}(\phi_{old}(\boldsymbol{x}_j), \phi_{new}(\boldsymbol{x}_q)). \tag{2}$$

## 3 RELATED WORK

As Li et al. (2015) realised, different models do not always generate compatible features, even when they are trained on the same dataset. This observation has motivated the rise of the relatively new literature on model compatibility: given a weaker old model $\phi_{old}$ we aim

to train a new model $\phi_{new}$ that has better retrieval performance, as in (1), whilst still is compatible with the old model, as in (2). This way we can improve performance, without having to wait months and incur large computational cost to backfill the gallery. Shen et al. (2020) introduced Backwards Compatible Training (BCT). They add an influence loss term to the new model objective that runs the old model linear classifier on the features generated by the new model to encourage compatibility. Budnik & Avrithis (2021) train a new model by formulating backward-compatible regularization as a contrastive loss. Zhang et al. (2021) proposed a Regression-Alleviating Compatible Training (RACT) method that adds a regularization term to the contrastive loss in order to reduce *negative flips*: these are images that the old model classified correctly but the new model gets wrong. Ramanujan et al. (2022) proposed the Forward Compatible Training (FCT) method that trains an independent new model first and subsequently trains a transformation function which maps features from old model embedding space to that of the new model. Moreover, the transformation requires some side-information about each data point as input. Iscen et al. (2020), Wang et al. (2020) and Su et al. (2022) also approach the model compatibility problem by learning maps between old and new model embedding spaces. Note that, as discussed in Ramanujan et al. (2022) the cost of applying the transformation is negligible compared to running the embedding model on images (hence, the instant jump in performance in Figure 1-right).

**Learning Compatible Predictions.** A related line of research focuses on learning compatible predictions (as opposed to compatible features). For the classification task, Srivastava et al. (2020) show that changing the random seed can lead to high prediction incompatibility even when the rest of the learning setup stays constant. Moreover, they observed incompatible points do not always have low confidence scores. Oesterling & Sapiro (2021) aim to achieve better prediction compatible models by ensuring that the new model training loss is smaller than the old model loss on all training samples. Träuble et al. (2021) consider the problem of assigning labels to a large unlabelled dataset. In particular, they focus on how to determine which points to re-classify when given a new model and how to proceed if the old and new predictions differ from each other. Similar to compatible representation learning, various works in the literature focus on positive-congruent training which aims to reduce the number of negative flips in the prediction setting. Yan et al. (2021) achieve this by focal distillation which puts a greater emphasis on samples that the old model classifies correctly. Zhao et al. (2022) argue that most negative flips are caused by logits with large margins and thus try to reduce large logit displacement.

**Continual Learning (CL) and Knowledge Distillation (KD).** Other less relevant areas of research that focus on learning across domains and tasks include CL and KD. CL (Parisi et al., 2019; Li & Hoiem, 2017; Zenke et al., 2017) is related to compatible learning as it also aims to train a new model that should keep the knowledge acquired by the old one. However, in the continual learning setup the new model is dealing with a new task, whereas here the task (image retrieval) is fixed. Also, in CL the two models never interact as they are not employed together: as soon as the new model is trained we discard the old one.

In KD (Hinton et al., 2015; Gou et al., 2021) we also train a new model (the student) that inherits some of the capabilities of an old model (the teacher). However, there are two main differences to model compatibility setup. Firstly, in KD the teacher model is often a larger and stronger model than the student, reverse of the setup in model compatibility. Further, the teacher and the student are never used together in practice. Therefore, the features of the two models do not need to be compatible in the KD setting. Zhang et al. (2021) show that neither CL methods nor KD ones are able to achieve good retrieval performance in the model update setting as the features of the old and new models are not compatible.

## 4 METHOD

### 4.1 FEATURE ALIGNMENT

Some previous works aim to achieve model compatibility when training the new model, by directly enforcing new features to be aligned to the existing old model features (Shen et al., 2020; Budnik & Avrithis, 2021; Zhang et al., 2021). However, this often greatly reduces the

single model retrieval performance defined in Eq. (1) for the new model, as they suffer from the inherent trade-off between maximizing the new model performance and at the same time its compatibility to the weaker old model. Other works have thus proposed learning a separate mapping from old model embedding space to new model embedding space to align their features (Wang et al., 2020; Meng et al., 2021; Ramanujan et al., 2022). Wang et al. (2020) and Meng et al. (2021) both still modify the training of the new model which is undesirable because it can limit the performance gain of the new model and is intrusive and complicated for practitioners. Ramanujan et al. (2022) train the new model and alignment map independently, but to achieve better alignment rely on existence of side-information (an extra feature generated by an unsupervised model computed for each data point). This is also undesirable because is not applicable to an already existing system without side-information available. We provide results in the presence of such side-information in the Appendix B.2. Here, as a baseline we consider the same feature alignment as in Ramanujan et al. (2022), but without side-information: we learn a map $h_\theta : \mathbb{R}^d \to \mathbb{R}^d$ from old model feature space to new model feature space modeled by an MLP architecture with parameters $\theta$. Hence, for cross-model retrieval we use $h_\theta(\phi_{old}(\boldsymbol{x}_j))$ instead of $\phi_{old}(\boldsymbol{x}_j)$ in Eq. (2).

## 4.2 FASTFILL

Despite relative success, all previous methods have limitations and fail to reach full-compatibility: in each case cross-model retrieval defined in Eq. (2) has significantly worse performance than single-model retrieval defined in Eq. (1) with the new model. In order to bridge this gap, (partial) backfilling is required: continuously move from cross model retrieval to single model retrieval by updating the old gallery features with the new model. That is to replace transformed old features $h_\theta(\phi_{old}(\boldsymbol{x}_i))$ with new features $\phi_{new}(\boldsymbol{x}_i)$ for some or all images in the gallery set ($\boldsymbol{x}_i \in \mathcal{G}$) (Figure 1); To this end, we propose our new method FastFill, which includes two main contributions: firstly, we change the backfilling curve behaviour for a random policy by modifying the alignment model training and secondly, we introduce a better policy that produces the order of samples to update, therefore results in promptly closing the gap between cross-model and single-model retrieval.

We propose a training loss for the alignment map $h$ to have both high retrieval performance at no backfilling (the instant jump after model update in Figure 1-right) and good backfilling curve, ensuring that we close the gap with new model performance promptly. To achieve this, we include objectives to encourage point-to-point and point-to-cluster alignments.

The first part of our training loss is the pairwise squared distance between the new features and the transformed old features: $\mathcal{L}_{l_2}(\boldsymbol{x}; h_\theta, \phi_{old}, \phi_{new}) = \|\phi_{new}(\boldsymbol{x}) - h_\theta(\phi_{old}(\boldsymbol{x}))\|_2^2$ for $\boldsymbol{x} \in \mathcal{D}_{new}$. This is the loss used by FCT (Ramanujan et al., 2022). Further, given $\phi_{old}$ and the new model classifier head, $\kappa_{new}$, that the new model has been trained on, we use a discriminative loss that runs $\kappa_{new}$ on the transformed features: $\mathcal{L}_{disc}(\boldsymbol{x}, y; h_\theta, \phi_{old}, \kappa_{new})$ where $(\boldsymbol{x}, y) \in \mathcal{D}_{new}$ is a pair of (image, label). $\mathcal{L}_{disc}$ is the same discriminative loss that new model is trained on: in our experiments it is the Cross Entropy loss with Label Smoothing (Szegedy et al., 2016) for ImageNet-1k and Places-365 datasets, and the ArcFace loss (Deng et al., 2019) for VGGFace2 dataset. This loss can be thought of as the reverse of the influence loss in BCT (Shen et al., 2020) but with one major advantage. Unlike BCT, we do not require old and new models to have been trained on the same classes as we are using the new model classifier head rather than the old one. Our proposed alignment loss is a combination of the two losses:

$$\mathcal{L}_{l_2+disc}(h_\theta; \phi_{old}, \phi_{new}, \kappa_{new}, \boldsymbol{x}, y) = \mathcal{L}_{l_2} + \mathcal{L}_{disc}. \tag{3}$$

$\mathcal{L}_{l_2}$ achieves pairwise compatibility, aligning each transformed old feature with the corresponding new feature. However, in practice we cannot obtain a perfect alignment on unseen data in the query and gallery sets. With $\mathcal{L}_{l_2}$ we encourage to reduce the magnitude of pair-wise alignment error $\phi_{new}(\boldsymbol{x}_i) - h_\theta(\phi_{old}(\boldsymbol{x}_i))$, but the direction of error is free. $\mathcal{L}_{disc}$ encourages the error vectors to be oriented towards the same-class clusters of the new model (given by $\kappa_{new}$) and away from wrong clusters. Hence, we avoid proximity of a cluster of transformed old features to a wrong cluster of new features – the mechanism responsible of slow improvements in backfilling.

We now focus on coming up with good ordering to backfill the images in the gallery set (replacing $h_\theta(\phi_{old}(\boldsymbol{x}_i))$'s with $\phi_{new}(\boldsymbol{x}_i)$'s). Finding *the optimal* ordering for backfilling is a hard combinatorial problem and infeasible. We first look at a heuristically good (but cheating) ordering: backfill the worst gallery features first. As minimizing the alignment loss (3) leads to good transformation function, we use it as a way to measure the quality of the gallery features: we compute the training loss (3) for each image in the gallery set and sort the images in decreasing order. In other words, we backfill the gallery images with the highest loss first as they are probably not very compatible with the new model. We call this method FastFill-Cheating. We show in Figure 2a that using FastFill-Cheating results in significantly better backfilling curve than when we backfill using a random ordering.

Unfortunately, we cannot use this backfilling method in practice for images in the gallery as we do not have access to $\phi_{new}(\boldsymbol{x}_i)$ until we have backfilled the $i$th image. Instead, we aim to estimate this ordering borrowing ideas from Bayesian Deep Learning. When the training objective is $\mathcal{L}_{l_2}$, we can model the alignment error by a multivariate Gaussian distribution with $\sigma\mathbb{I}$ covariance matrix[2], and reduce the negative log likelihood during training. This is similar to the method proposed by Kendall & Gal (2017). Given the old and new models, we train a model $\psi_\vartheta : \mathbb{R}^d \mapsto \mathbb{R}$ parameterized by $\vartheta$ that takes the old model feature $h(\phi_{old})$ as input and estimate $\log \sigma^2$. We can then jointly train $h_\theta$ and $\psi_\vartheta$ by minimizing:

$$\mathbb{E}_{\boldsymbol{x}\sim\mathcal{D}_{new}}\left[\frac{\|h_\theta(\phi_{old}(\boldsymbol{x})) - \phi_{new}(\boldsymbol{x})\|_2^2}{\sigma^2} + \frac{1}{\lambda}\log\sigma^2\right], \qquad (4)$$

where $\log\sigma^2 = \psi_\vartheta(\phi_{old}(\boldsymbol{x}))$, and $\lambda$ is a hyper-parameter function of $d$. Now, we extend this formulation for our multi-objective training in (3). We follow the approximation in Kendall et al. (2018) for uncertainty estimation in a multi-task setup where the model has both a discrete and continuous output. Further, we assume the same covariance parameter, $\sigma^2$, for both of our regression and discriminative objectives ($\mathcal{L}_{l2}$ and $\mathcal{L}_{disc}$). Hence, we can replace $\mathcal{L}_{l_2}$ in (4) with $\mathcal{L}_{l_2+disc}$ and get the full FastFill training loss:

$$\mathcal{L}(h_\theta, \psi_\vartheta; \phi_{old}, \phi_{new}, \kappa_{new}\mathcal{D}_{new}) = \mathbb{E}_{(\boldsymbol{x},y)\sim\mathcal{D}_{new}}\left[\frac{\mathcal{L}_{l_2+disc}}{\sigma^2} + \frac{1}{\lambda}\log\sigma^2\right]. \qquad (5)$$

In practice, we use a shared backbone for $h_\theta$ and $\psi_\vartheta$, therefore the overhead is minimal. Please see Appendix A.2 for details of architecture, hyper-parameters, and training setup. In Figure 2b we show that the predicted $\sigma_i^2$ by $\psi_\vartheta$ trained with (5) is a good approximation of $\mathcal{L}_{l_2+disc}(\boldsymbol{x}_i)$ for $\boldsymbol{x}_i \in \mathcal{G}$. In particular, the ordering implied by $\mathcal{L}_{l_2+disc}$ and $\sigma$ are close. In Figure 2c we show the order of each $\boldsymbol{x}_i$ in the gallery when sorted by $\mathcal{L}_{l_2+disc}(\boldsymbol{x}_i)$ is well correlated with its order when sorted by $\sigma_i^2$: The Kendall-Tau correlation between two orderings is 0.67 on the ImageNet test set. In FastFill, we backfill $\boldsymbol{x}_i$'s with the ordering implied by their predicted $\sigma_i^2$'s (from high to low values).

Finally, in Figure 2a we show that FastFill obtains similar backfilling curve as the cheating setup using the ordering implied by cheaply computed $\sigma_i^2$'s. Also note that, FastFill even with a random ordering obtains significant improvement compared to our baseline (FCT with random ordering) due to superiority of our proposed alignment loss. In Section 6 we further demonstrate the importance of our proposed training objective and backfilling ordering.

## 5 Experiments

We now analyse the performance of our method by comparing it against different baselines on a variety of datasets. We first describe the metrics we use to evaluate and compare different compatibility and backfilling methods.

### 5.1 Evaluation Metrics

**Compatibility Metrics** Similar to previous model compatibility works we mainly use following two metrics to evaluate the performance and compatibility of models. The *Cumulative Matching Characteristics (CMC)* evaluates the top-$k$ retrieval accuracy. We compute

---

[2]We also considered an arbitrary diagonal covariance matrix instead of a scaled identity matrix, and observed no significant improvement.

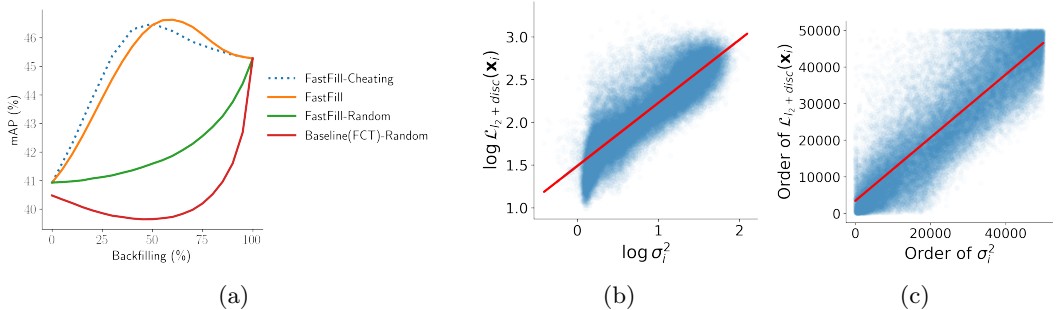

(a)         (b)         (c)

Figure 2: a) Backfilling results for different training objectives and backfilling orderings evaluated on the ImageNet-1k dataset. The FastFill training objective, defined in Eqn. (5), compared to our baseline method (FCT) obtains significantly improved mAP over the course of backfilling when a random ordering is used. The performance is further improved when using the ordering implied by the predicted uncertainties $\sigma_i^2$. We also compare against FastFill-Cheating, in which the ordering is obtained by computing the training loss (3) on gallery images (hence a cheating setup), and show comparable performance. b) We show that the predicted $\log \sigma_i^2$, the output of $\psi_\vartheta$ on gallery images, and $\log \mathcal{L}_{l_2+disc}(\boldsymbol{x}_i)$ are well correlated. c) We sort $\boldsymbol{x}_i$ once by $\sigma_i^2$ and once by $\mathcal{L}_{l_2+disc}(\boldsymbol{x}_i)$ to get two orderings. Here, for each $\boldsymbol{x}_i$ we plot its order in the first ordering vs its order in the second ordering, and observe great correlation (Kendall-Tau correlation=0.67).

the distance between a single query feature and every single gallery features. We then return the $k$ gallery features with the smallest distance. If at least one of the $k$ returned gallery images has the same label as the query we consider the retrieval to be successful. We also compute the *mean Average Precision (mAP)* score, which calculates the area under the precision-recall curve over recall values between 0.0 and 1.0.

Given a metric $M$ (either top-$k$ CMC or mAP), a gallery set $\mathcal{G}$, and query set $\mathcal{Q}$ we denote the retrieval performance by $M(\mathcal{G}, \mathcal{Q})$. After a model update, we use $\phi_{new}$ to compute features for the query set and $\phi_{old}$ to compute those for the gallery set. Similar to previous works, for retrieval metrics, we use the full validation/test set of each dataset as both the query set and gallery set. When we perform a query with an image, we remove it from the gallery set to avoid a trivial match.

**Backfilling Metrics** In partial backfilling we replace some of the old features, here $h(\phi_{old}(\boldsymbol{x}_i)))$, from the gallery with new features, $\phi_{new}(\boldsymbol{x}_i)$. In order to evaluate the retrieval performance for a partially backfilled gallery we introduce the following notation: given an ordering $\pi : \boldsymbol{x}_{\pi_1}, \ldots, \boldsymbol{x}_{\pi_n}$ of the images in the gallery set and a backfilling fraction $\alpha \in [0, 1]$, we define $\mathcal{G}_{\pi,\alpha}$ to be the partially backfilled gallery set. We use the new model to compute features for $\boldsymbol{x}_i \in \pi[: n_\alpha]$ and the old model for the remaining images, that is for all $\boldsymbol{x}_i \in \pi[n_\alpha :]$, using numpy array notation, where $n_\alpha = \lfloor \alpha n \rfloor$ and $n$ is the size of gallery.

Given a metric $M$, to compare different backfilling and model update strategies during the entire course of model update, we propose the following averaged quantity:

$$\widetilde{M}(\mathcal{G}, \mathcal{Q}, \pi) = \mathbb{E}_{\alpha \sim [0,1]} M(\mathcal{G}_{\pi,\alpha}, \mathcal{Q}) \tag{6}$$

$M$ can be any retrieval metric such as top-$k$ CMC or mAP as defined above. The backfilling metric $\widetilde{M}$ corresponds to the area under the backfilling curve for evaluation with $M$.

## 5.2 BACKFILLING EXPERIMENTS

We now analyse our method by comparing it to three baselines: BCT, RACT, and FCT. For BCT and FCT we used the official implementation provided by the authors. We re-implemented RACT and optimized over the hyper-parameters as the code was not publicly available at the time of writing. We ran all experiments five times with different random seeds and report the averaged results. We use the $l_2$ distance for image retrieval.

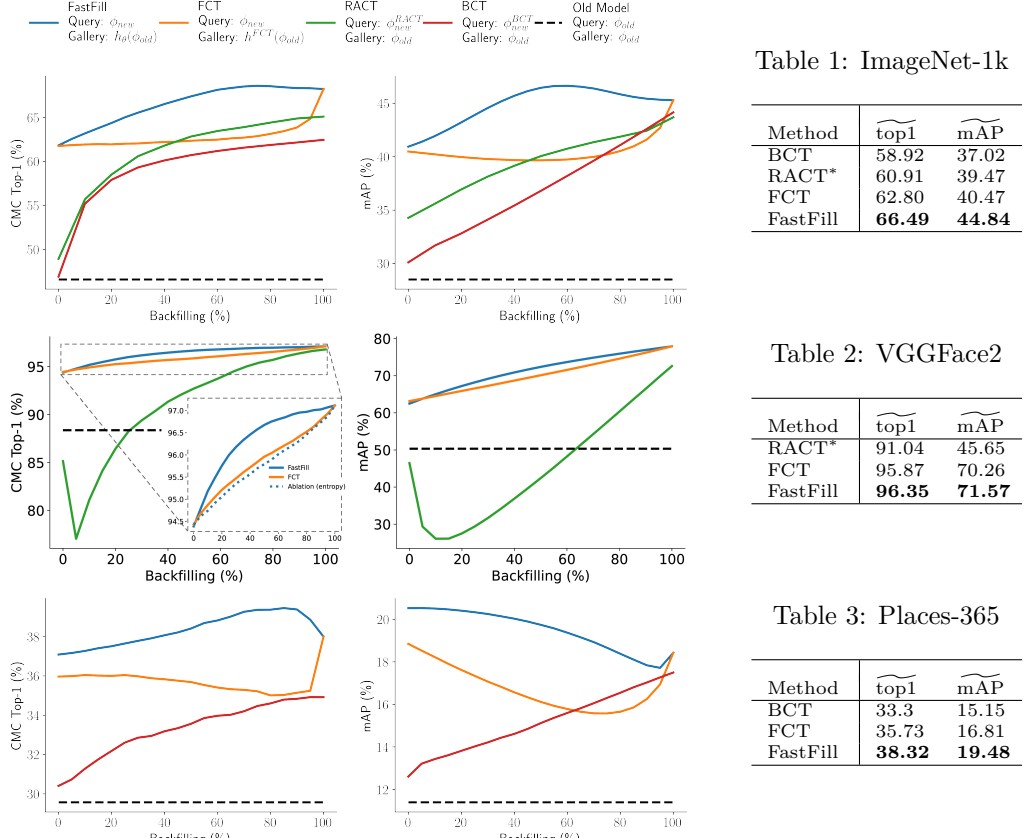

Table 1: ImageNet-1k

| Method | $\widetilde{\text{top1}}$ | $\widetilde{\text{mAP}}$ |
|---|---|---|
| BCT | 58.92 | 37.02 |
| RACT* | 60.91 | 39.47 |
| FCT | 62.80 | 40.47 |
| FastFill | **66.49** | **44.84** |

Table 2: VGGFace2

| Method | $\widetilde{\text{top1}}$ | $\widetilde{\text{mAP}}$ |
|---|---|---|
| RACT* | 91.04 | 45.65 |
| FCT | 95.87 | 70.26 |
| FastFill | **96.35** | **71.57** |

Table 3: Places-365

| Method | $\widetilde{\text{top1}}$ | $\widetilde{\text{mAP}}$ |
|---|---|---|
| BCT | 33.3 | 15.15 |
| FCT | 35.73 | 16.81 |
| FastFill | **38.32** | **19.48** |

Figure 3: Backfilling results on ImageNet-1k (top), VGGFace2 (middle), and Places-365 (bottom). We use new model features for the query set. For the gallery set we start off using (transformed) old features and incrementally replace them with new features.

**ImageNet-1k.** We compare FastFill against BCT, RACT, and FCT, three state-of-the-art methods. For RACT we use entropy-backfilling; since BCT and FCT do not offer a policy for backfilling we use random orders. FastFill outperforms all three methods by over 3.5% for $\widetilde{\text{CMC-top1}}$ and over 4% for $\widetilde{\text{mAP}}$ (Figure 3, Table 1). Compared to BCT and RACT, FastFill performs significantly better for 0% backfilling (+10%) due to the transformation function. Furthermore, the sigma-based ordering significantly outperforms the random orderings and the cross-entropy one from RACT. We show in Appendix D that FastFill both increases the number of positive flips and decreases the negative flips compared to FCT.

**VGGFace2.** We show results on the VGGFace2 dataset in Table 2 and Figure 3. FastFill improves the baselines as well as an ablation that combines entropy-based backfilling with the FastFill transformation by 0.5% $\widetilde{\text{CMC-top1}}$ and 1.3% $\widetilde{\text{mAP}}$. The absolute improvement is relatively small as the gap between 0% and 100% backfilling is not very large, as the transformation already achieves high compatibility.

**Places-365.** We run experiments on the Places-365 dataset. The results are summarized in Table 3 and Figure 3. FastFill beats FCT with random backfilling by about 2.7% for both metrics. We further report CMC-top5 results in Appendix B.3 for all three datasets.

### 5.3 UPDATING BIASED MODELS

Embedding models can be biased toward certain sub-groups in their input domain, a crucial problem in image retrieval. For example, in face recognition tasks the model

can be biased toward a certain ethnicity, gender, or face attribute. It is often due to a class-imbalanced training dataset. Here, we show FastFill can fix model biases efficiently without requiring expensive full backfilling. We train a biased old model on the VGGFace2-Gender-550 dataset that consists of 50 female and 500 males classes. The old model performs significantly better on males identities (90.6%) in the test set than females (68.7%).

We train a new model on a larger gender-balanced dataset that consists of 2183 female and 2183 male identities. The new model still has a small bias towards male classes (98.3% vs 97.1%). Applying the feature alignment trained with FastFill alleviates some of the biases reducing the accuracy gap between genders to 9.4%. As we show in Figure 4, with only 25% back-filling FastFill reaches the same accuracy gap between genders as the new model, whereas random backfilling using FCT requires backfilling the entire gallery set to reduce the bias to the level of the new model. FastFill prioritizes backfilling instances corresponding to female identities which results in a fast reduction in the gender gap compared to previous methods (see Appendix F).

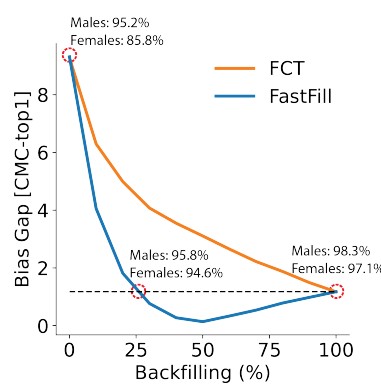

Figure 4

## 6 ABLATION STUDY

Here, we perform an ablation study to highlight the different components of our method (Figure 5). We first compare the effect of different training losses ($\mathcal{L}_{l_2}$, $\mathcal{L}_{disc}$, $\mathcal{L}_{l_2+disc}$ with and without uncertainty) when using random backfilling orderings. We show that $\mathcal{L}_{l_2+disc}$ greatly improves the backfilling curve. Training with uncertainty has little effect on random backfilling. We also show that using $\sigma$ for the backfilling ordering greatly improves performance. We conclude that FastFill (training with $\mathcal{L}_{l_2+disc}$ and uncertainty) obtains the best results for both random and $\sigma$-based orderings. We show in Appendices C.3 and C.4 that FastFill is particularly strong when we encounter new classes at inference time as the relative confidence (used in RACT) between samples from new classes are not as informative.

## 7 CONCLUSION

Model compatibility is a crucial problem in many large-scale retrieval systems and a common blocker to update embedding models. In this work, we propose FastFill: an online model update strategy to efficiently close the accuracy gap of previous model compatibility works.

The key idea in FastFill is to perform partial backfill-ing: using a new model we recompute a small fraction of the images from a large gallery set. To make the partial backfilling process efficient we propose a new training objective for the alignment model which maps the old model features to those of the new model, and an ordering of the gallery images to perform backfill-ing. To obtain the ordering we train the alignment model using uncertainty estimation and use predicted uncertainties to order the images in the gallery set.

We demonstrate superior performance compared to previous works on several datasets. More importantly, we show that using FastFill a biased old model can be efficiently fixed without requiring full backfilling.

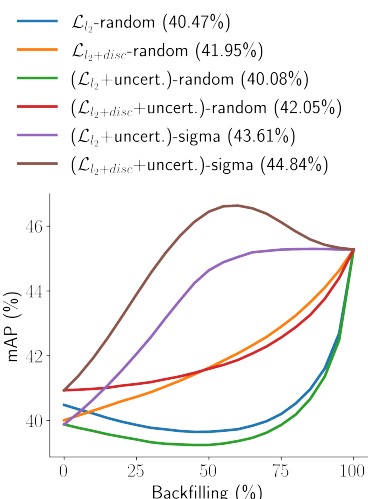

Figure 5: Ablation Study on Ima-geNet: We compare the effect of dif-ferent training losses and backfilling orderings on the backfilling curves. We report $\widetilde{mAP}$ in the parentheses in the legend.

## 8 ETHICS STATEMENT

We believe that FastFill can have a positive impact in real-world applications by reducing the cost associated with model update, and hence enable frequent model updates.

Machine learning based models are often not completely unbiased and require regular updates when we become aware of a certain limitation and/or bias. As shown in the paper, when we have a new model with fewer biases than an existing model, FastFill can reach the fairness level of the new model more quickly and cheaply. This can motivate frequent model updates to address limitations associated with under represented groups.

As with many machine learning applications, especially ones used for facial recognition, FastFill can be misused. Facial recognition software can be used in many unethical settings, so improving their performance can have significant downsides as well as upsides.

## 9 LIMITATIONS

Our proposed method does not provide a mechanism to decide what percentage of backfilling is sufficient to enable early stopping. Knowing how much backfilling is sufficient would result in saving additional computation.

In some real world applications, the image representations can be used in a more sophisticated fashion by the downstream task than the nearest neighbour based retrieval considered in this work. For instance, representations can be input to another trained model. Therefore, when we update from old to new, we also need to retrain the downstream model so it is compatible with new feature space. This is not a limitation of FastFill in particular, however, in general a limitation of transformation based update methods.

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

# A   EXPERIMENTAL SET-UP

## A.1   DATASET SET-UP

**ImageNet-1k (21).**   ImageNet-1k is a large-scale image recognition dataset and the most used subset of ImageNet [ILSVRC]. It was used in the ILSVRC 2012 image recognition challenge. The dataset contains 1000 object classes and a total of 1,281,167 training images and 50,000 validation images. The training dataset is relatively balanced with about 1.2k images per class. The validation set has the same classes as the training set and exactly 500 images per class.

We use a subset of ImageNet-1k consisting of the first 500 classes to train the old model; we refer to this as ImageNet-500. We note that the first half of ImageNet-1k contains easier classes than the second half.

**Places-365 (32).**   Places-365 is a large-scale scene recognition dataset that spans 365 classes and contains 1.8 million training images and 36,500 validation ones. The training set contains between 3068 and 5000 images per category and the validation set has exactly 100 images per class. We use the first 182 classes of the training set to create a smaller subset; we refer to it as Places-182.

**VGGFace2 (2).**   VGGFace2 is a large-scale facial recognition dataset. Its training dataset is made up of 8631 classes, each representing a different person, and a total of 3.14 million images. The test set contains 170,000 images corresponding to 500 identities, different to the ones present in the training set. The training set is slightly unbalanced with an average of 362.6, a minimum of 57, and a maximum of 843 images per class. The test set contains exactly 500 images per class. During test time, we compute a random but fixed subsample of 50 images per class in the test set.

The dataset also has a list of attributes for a small number of images in the training dataset. We know the gender of 30,000 images corresponding to 5300 different identities. We create a training and validation set of images for which we know the gender as follows: we disregard all the images in the training set with an unknown gender. We randomly sample 250 female identities and 250 male ones to form the new validation set. As there are more male than female classes remaining, we sample a subset of the remaining male identities to create a balanced training dataset (2183 female and male classes each with about 800,000 images for either (exact numbers depend on the random seed)). We refer to this new subset as **VGGFace2-Gender**. We now create an unbalanced subset by sampling 500 male identities

and 50 female identities from the training set, we name it VGGFace2-Gender-550. We use this dataset to train a biased old model.

## A.2 Training Set-Up

**ImageNet-1k.** We train two ResNet50 models ([7]): the old model ($\phi_{old}$) on ImageNet-500, and the new model ($\phi_{new}$), on ImageNet-1k. We set the embedding dimension of both models to 128. We train both models using the hyper-parameters in ResNetV1.5 Nvidia ([17]) (SGD optimizer, epochs=100, batchsize=1024, learning rate=1.024, weight decay=$3.0517578125 * 10^{-5}$, momentum=0.875, cosine learning rate decay with 5 epochs of linear warmup). For our method, we jointly train a transformation model $h$, and an uncertainty predictor $\psi$ on ImageNet-1k using the objective function defined above ([5]). We use the same architecture for the MLP architecture for the transformation function $h$ as ([20]) and the same learning set-up (Adam Kingma & Ba ([13]), epochs=80, learning rate=$5 * 10^{-4}$, cosine learning rate decay with 5 epochs of linear warm-up, freeze of the BatchNorm layers after 40 epochs). For the uncertainty estimator $\psi$, we simply use a linear layer from the transformed embeddings space to output a single real value. For FCT, we train their transformation without side-information exactly as described in their paper and their official code base (`https://github.com/apple/ml-fct`). For BCT we train a new model using the official code implementation (`https://github.com/YantaoShen/openBCT`) and hyper-parameters mentioned in the paper. However, we set the embedding dimension to 128 to be directly comparable to the other three models. For RACT, we re-implement the method based on the information provided in the paper as there is no official code release. We set the hyper-parameters ($\lambda = 1$ and $\tau = 0.5$) as defined in the paper and optimize over the learning rate to get optimal performance. We compare the learning rate schedule used in RACT with the one used in ResNet50V1.5 Nvidia ([17]). optimizing over the learning rate for each (we try learning rates in powers of ten in [1e-5, 1]).

For the retrieval experiments we set both the gallery and query sets to be the validation set of ImageNet-1k. For BCT and FCT we use random backfilling, as neither paper include a backfilling ordering. For RACT we try both random backfilling and entropy backfilling and pick the better one.

**Places-365.** We use a similar setup to ImageNet experiments: we train two ResNet50 models with an embedding size of 512: the old model on Places-182 and the new one on Places-365. We use the Places-365 validation dataset for retrieval evaluation.

**VGGFace2.** For the VGGFace2 dataset we train a smaller ResNet18 model on VGGFace2-863 and a larger ResNet50 one on VGGFace2-8631. Unlike for the other two datasets we use the ArcFace objective ([4]) to train the two models and the transformation. We use a margin of 0.5 and a scale of 64 as proposed by ([4]). We use the same learning rate schedule as for ImageNet. Unlike for the other two datasets we normalize the features.

# B    Further Experimental Results

## B.1    Architectural Change Experiments

We analyse how FastFill compares to existing methods when the old and new models have different architectures. We run ImageNet experiments where the old model is a ResNet-18, and the new model a ResNet-50. Both are trained on the same full ImageNet-1k training set. As before, we repeat the experiments with 5 random seeds and illustrate the results in Table 4 and Figure 6. The overall trends are the same as the other setups in the original paper. FastFill significantly outperforms all baselines.

Table 4: ImageNet-1k Experiments on architectural changes: The old model is a ResNet-18 and the new model a ResNet-50.

| Method | $\widetilde{\text{top1}}$ | $\widetilde{\text{mAP}}$ |
|---|---|---|
| Old model | 54.04 | 26.41 |
| FastFill | **67.77** | **45.94** |
| FCT | 64.91 | 43.43 |
| RACT | 64.80 | 42.04 |
| BCT | 59.26 | 36.56 |

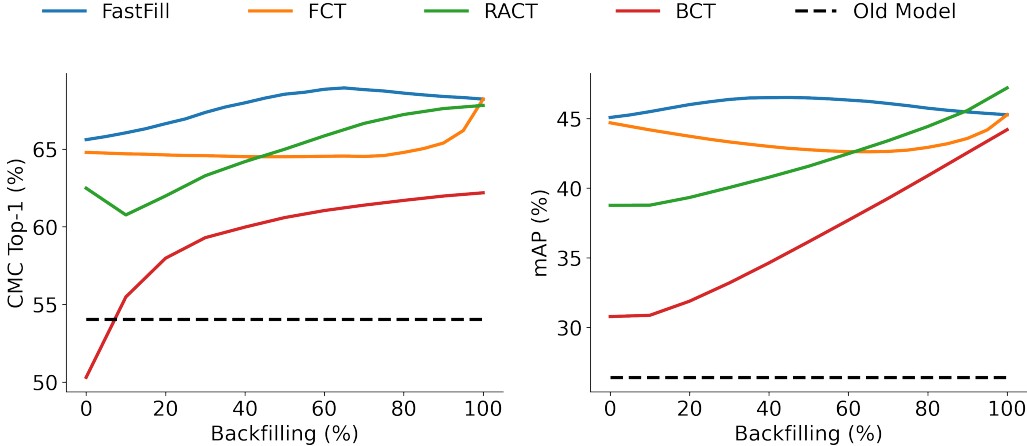

Figure 6: We run experiments on ImageNet to investigate the performance of an architecture change. The old model is a ResNet-18 and the new model a ResNet-50; both are trained on ImageNet-1k.

## B.2 SIDE-INFORMATION EXPERIMENTS

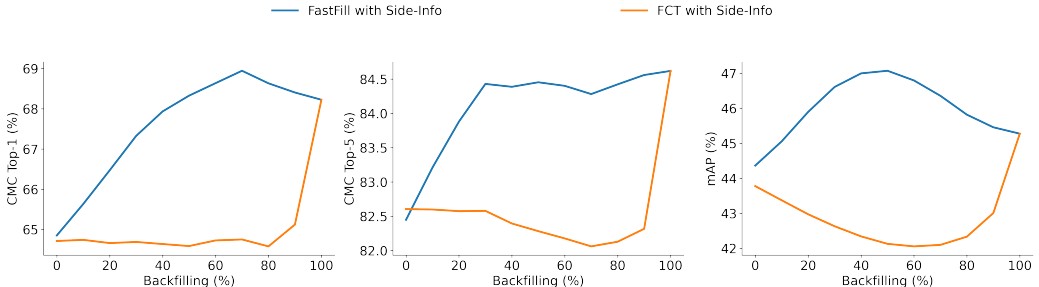

Figure 7: We add side-information to the transformation function trained with either FastFill (blue) or FCT (orange). FastFill outperforms FCT on all three metrics: CMC-top1 (left), CMC-top5 (middle), and mAP (right).

We now train both the FCT transformation model ($h$) as well as the FastFill transformation and uncertainty models ($h$ and $\psi$) with side-information. For the side-information model we train a ResNet50 model on ImageNet-500 with SimCLR (3). Otherwise, we use the same set-up as for the main paper ImageNet experiments. We show in Figure 7 that both methods get a performance boost with side-information. But FastFill still significantly outperforms FCT on three different metrics: $\widetilde{\text{CMC}}$-top1 (67.81% vs 64.72%), $\widetilde{\text{CMC}}$-top5 (84.23% vs 82.35%), and $\widetilde{\text{mAP}}$ (46.23% vs 42.55%).

## B.3 CMC-TOP5

We run experiments using the CMC-top5 retrieval metric. As shown Figure 8 FastFill outperforms all baselines on ImageNet-1k (+2.8%), VGGFace2 (0.1%), and Places-365 (+2.1%) when using CMC-top5 to evaluate retrieval performance.

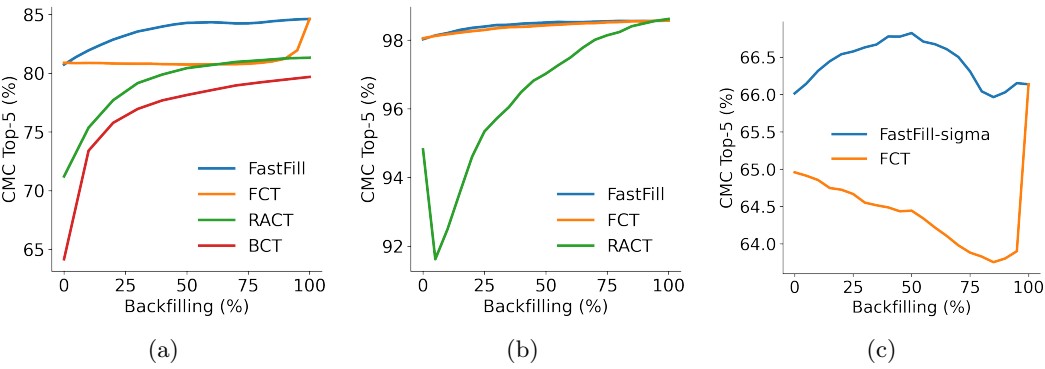

Figure 8: We run experiments using the CMC-top5 retrieval metric on (a) ImageNet-1k, (b) VGGFace2, and (c) Places-365.

## C  Further Ablation Studies

### C.1  Ablation on ImageNet-1k

We elaborate from the ablation study provided in the main paper in section 6 in Figure 9 and Table 5 adding the CMC-top1 metric. In particular we show that when training with $\mathcal{L}_{disc}$ with or without uncertainty we get poor compatibility performance at 0% backfilling and throughout the backfilling (when measured using mAP). However, adding it to $\mathcal{L}_{l_2}$ to get $\mathcal{L}_{l_2+disc}$ improves results for both random and $\sigma^2$ based backfilling for both mAP and CMC-top1.

Table 5: Ablation Study: We compare the effect of different training losses and backfilling orderings on the backfilling curves.

| Method | $\widetilde{\text{top1}}$ | $\widetilde{\text{mAP}}$ |
|---|---|---|
| $(\mathcal{L}_{l_2})$-random | 62.80 | 40.47 |
| $(\mathcal{L}_{disc})$-random | 65.51 | 39.45 |
| $(\mathcal{L}_{l_2+disc})$-random | 65.06 | 41.95 |
| $(\mathcal{L}_{l_2}+\text{uncert.})$-random | 62.58 | 40.08 |
| $(\mathcal{L}_{l_2+disc}+\text{uncert.})$-random | 64.51 | 42.05 |
| $(\mathcal{L}_{l_2}+\text{uncert.})$-sigma | 65.76 | 43.61 |
| $(\mathcal{L}_{disc}+\text{uncert.})$-sigma | 55.21 | 27.59 |
| $(\mathcal{L}_{l_2+disc}+\text{uncert.})$-sigma | **66.49** | **44.84** |

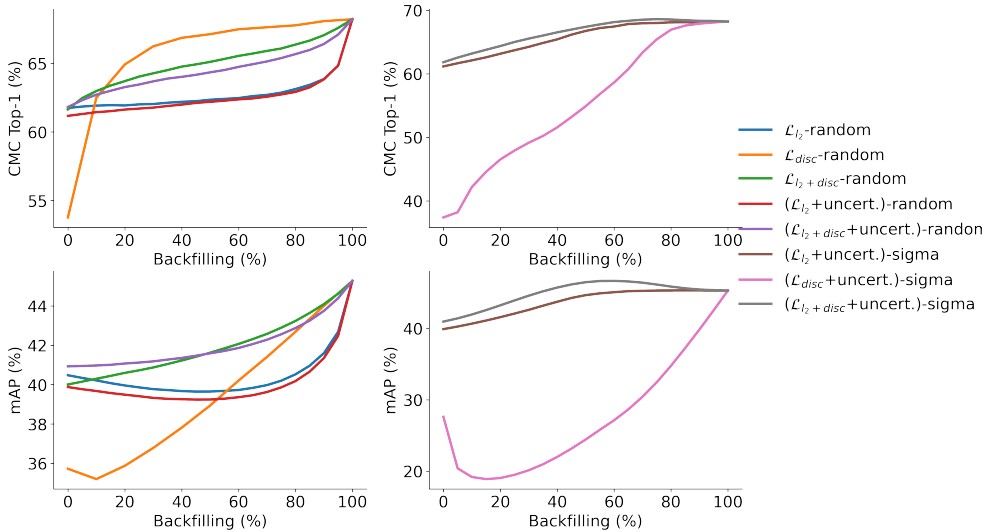

Figure 9: Ablation Study: We compare the effect of different training losses and backfilling orderings on the backfilling curves.

## C.2 ABLATION ON PLACES-365

We provide an ablation study in Figure 10 and Table 6 on the Places-365 dataset. The conclusion is similar to the ImageNet ablation where FastFill improves its ablations using alternative uncertainty measures and significantly improves previous work FCT.

Table 6: Ablation Study on Places-365: We compare the effect of different training losses and backfilling orderings on the backfilling curves.

| Method | Transformation | Uncertainty Estimation | $\widetilde{\text{top1}}$ | $\widetilde{\text{mAP}}$ |
|---|---|---|---|---|
| Old Model | ✗ | ✗ | 29.57 | 11.39 |
| FCT | ✓ | ✗ | 37.73 | 16.81 |
| FastFill | ✓ | Bayesian | **38.32** | **19.48** |
| FastFill-Ablation | ✓ | Entropy | 38.12 | 18.6 |

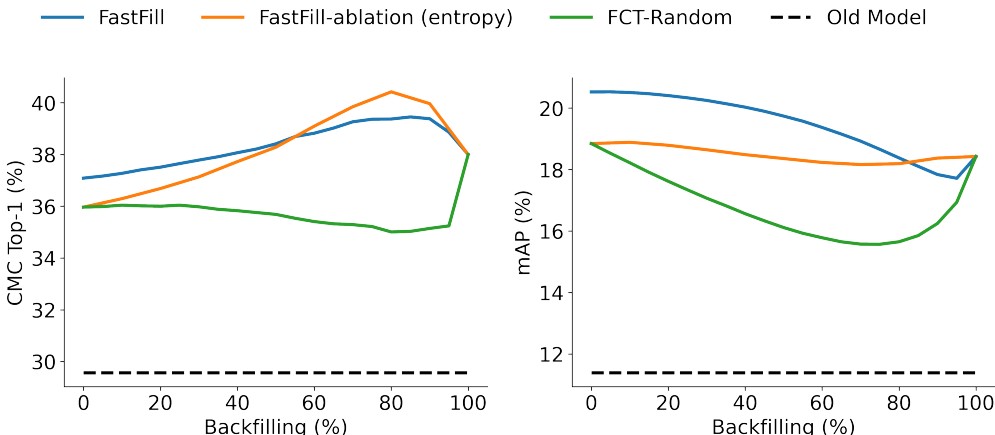

Figure 10: Ablation Study on Places-365: We compare the effect of different training losses and backfilling orderings on the backfilling curves.

## C.3 Ablation on ImageNet-250 to ImageNet-500

We provide a further ablation study that compares different evaluation set-ups focusing on different backfilling orderings provide by different uncertainty estimations. We run experiments where the same classes are used during training and evaluation as well as experiments where we evaluate the compatibility on previously unseen classes. The results are illustrated in Figure 11 and Table 7. In every case we train the old model on ImageNet-250 and the new model on ImageNet-500. For the first set of experiments ([500: ]) we use the first 500 classes for both the query and gallery. This is the **same train-test classes** setup. For the second experiment ([500:]) we use the second 500 classes for both query and gallery. This is the **disjoint train-test classes** setup. And finally we use all 1000 classes for both query and gallery ([:1000]). This is a setup where **half of the train-test classes** overlap.

The below results show that Bayesian estimation (FastFill) brings further improved accuracy in the presence of new classes. For example, compared to its ablations, FastFill obtains +1% top-1 improvement when evaluated on the first 500 classes [:500], but gets +2.7% top-1 improvement when evaluated on the second 500 classes [500:] that are unseen during training. Further, when evaluated on unseen classes, ablations of FastFill with logits-based uncertainty estimation gets results close to no uncertainty estimation at all as in FCT.

We note that for most real-world image retrieval tasks (as for instance facial recognition, or zero-shot retrieval) the test-time samples (for gallery and query sets) are coming from new classes that were not present at training-time. Using a training time classification head to measure uncertainty as in RACT (30) is a major limitation in practice since the relative confidence or entropy between the samples from new classes tend to be not as informative. In contrast, our proposed uncertainty estimation method based on feature alignment is explicitly designed to be class agnostic and works well on previously unseen classes. This is one of our method's main contributions and novelties.

Table 7: Ablation Study on ImageNet: We compare the effect of different training losses and backfilling orderings on the backfilling curves. The old model is trained on ImageNet-250 and the new one on ImageNet-500.

| Method | Transformation | Uncertainty Estimation | $\widetilde{\text{top1}}$ [: 500] | $\widetilde{\text{top1}}$ [500 :] | $\widetilde{\text{top1}}$ [: 1000] | $\widetilde{\text{mAP}}$ [: 500] | $\widetilde{\text{mAP}}$ [500 :] | $\widetilde{\text{mAP}}$ [: 1000] |
|---|---|---|---|---|---|---|---|---|
| Old Model | ✗ | ✗ | 50.7 | 12.41 | 29.32 | 29.13 | 2.47 | 14.9 |
| FastFill | ✓ | Bayesian | **75.27** | **17.9** | **43.0** | **53.88** | **4.68** | **26.44** |
| FastFill-ablation | ✓ | Entropy | 74.26 | 15.22 | 40.78 | 53.48 | 4.13 | 25.74 |
| FastFill-ablation | ✓ | Margin Conf. | 74.21 | 15.17 | 40.66 | 53.21 | 4.01 | 25.56 |
| FastFill-ablation | ✓ | Least Conf. | 74.28 | 15.18 | 40.74 | 53.38 | 4.07 | 25.65 |
| FCT | ✓ | ✗ | 71.15 | 14.94 | 38.77 | 48.61 | 3.7 | 23.4 |
| RACT | ✗ | Entropy | 66.88 | 13.65 | 36.65 | 41.6 | 3.3 | 20.4 |

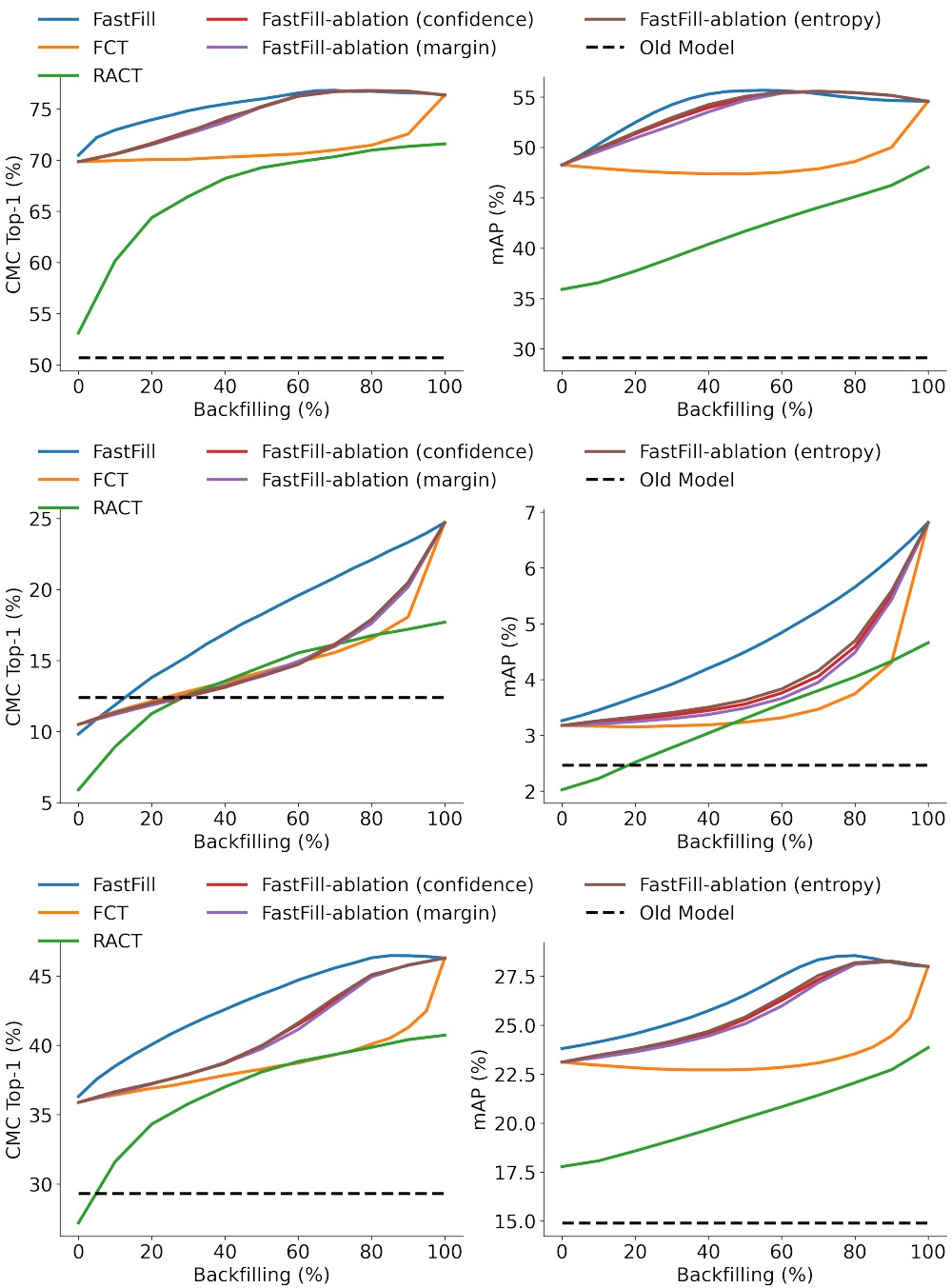

Figure 11: Ablation Study on ImageNet: We compare the effect of different training losses and backfilling orderings on the backfilling curves. **Top:** [500: ] Use the first 500 classes for both query and gallery. This is the same train-test classes setup. **Middle:** [500:] Use the second 500 classes for both query and gallery. This is the disjoint train-test classes setup. **Bottom:** [:1000] Use all 1000 classes for both query and gallery. This is a setup where half of the train-test classes overlap. These results show that FastFill is particularly strong when we encounter new previously unseen classes during inference time as is common in many real life retrieval settings

## C.4 ABLATION ON VGGFACE2

We run experiments on VGGFace2 using confidence and entropy based backfilling (as proposed by Zhang et al. (30)) on top of our proposed loss for feature alignment and the FCT baseline. We illustrate the results in Figure 12.

We observe that entropy/confidence based ablation on top of FastFill and baseline FCT are quite close to random backfilling whereas our proposed uncertainty estimation significantly improves them. Moreover, FastFill variants utilize a transformation function from old to new feature space that leads to significantly improved model update accuracies compared to RACT which directly enforce compatibility between new and old models (more than 5% on top-1 and 25% on mAP metrics).

We note that the uncertainty estimation measures proposed in Zhang et al. (30) (confidence and entropy) are intuitive and simple to use, but rely on the training time classification head. For most real-world image retrieval tasks (e.g. face recognition, or zero-shot retrieval) the test-time samples (for gallery and query sets) are coming from different classes to those seen during training-time. Using the training-time classification head to measure uncertainty as in Zhang et al. (30) is a major limitation in practice since the relative confidence/entropy between the samples from new classes tend to be not informative. In contrast, our proposed uncertainty estimation method based on feature alignment is explicitly designed to be class agnostic and works well on previously unseen classes. This is one of our method's main contributions/novelties.

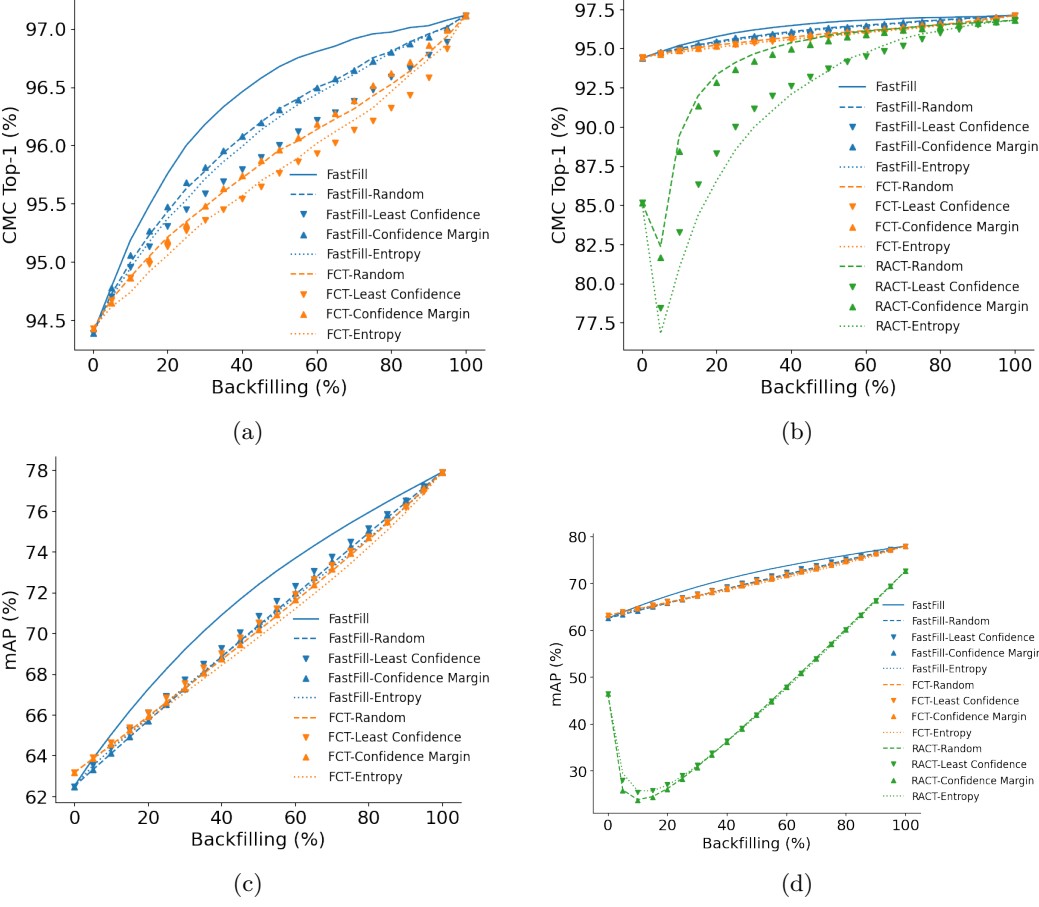

Figure 12: We run experiments on VGGFace2 using confidence and entropy based backfilling (as proposed by Zhang et al. (30)) on top of our proposed loss for feature alignment and the FCT baseline. We compare different methods using the CMC-top1 (a, b) and mAP (c, d) metrics, respectively. In (b) and (d) we further compare against RACT.

# D  POSITIVE AND NEGATIVE FLIPS ANALYSIS

We analyse the positive and negative flips for FastFill and FCT. A negative flip is an image that gets classified correctly for cross-model retrieval with 0% but gets misclassified when using $\alpha \in (0, 1]$ amount of backfilling. Conversely a positive flip is an image that originally gets misclassified but then labelled correctly for some amount of (partial) backfilling. The retrieval performance at a certain backfilling point is a function of the original cross-model retrieval performance at 0% backfilling and plus the number of positive minus the negative flips. We show FastFill outperforms FCT by both increasing the number of positive flips with limited backfilling whilst also causing fewer negative flips throughout the entire backfilling curve. We note that even though at 100% backfilling both FCT and FastFill use the same model, the number of flips doesn't match as it also depends on the cross model performance at 0% backfilling, which is different for the two methods.

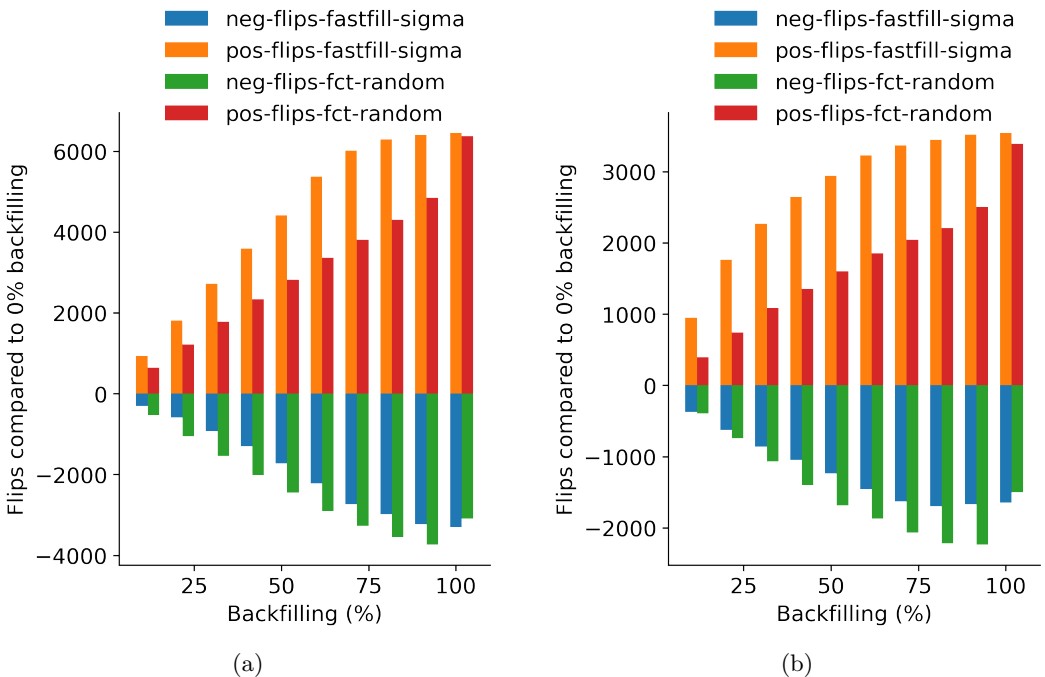

(a)  (b)

Figure 13: Positive and Negative Flip Analysis on ImageNet. We compare FastFill (blue and orange) and FCT (green and red) on ImageNet using (a) CMC-top1 and (b) top-5 retrieval, respectively. The aim is to maximize the number of positive flips (orange and red), whilst minimizing the number of negative ones (blue and green) with as little backfilling as possible.

# E  Sigma Analysis

Further to the analysis done in the main paper above, we show how well $\sigma^2$ correlates with different loss functions ($\mathcal{L}_{l_2+disc}(\boldsymbol{x}_i)$, $\mathcal{L}_{l_2}(\boldsymbol{x}_i)$, and $\mathcal{L}_{disc}(\boldsymbol{x}_i)$) and the orderings that they induce. Our prediction $\sigma^2$ most closely correlates with $\mathcal{L}_{l_2+disc}(\boldsymbol{x}_i)$ (the two orderings have a Kendall-Tau correlation of 0.67) which is unsurprising as it is trained to predict this loss. However, it still predicts both $\mathcal{L}_{l_2}(\boldsymbol{x}_i)$ (Kendall-Tau correlation of 0.65) and $\mathcal{L}_{disc}(\boldsymbol{x}_i)$ (Kendall-Tau correlation of 0.63) well.

When we train our uncertainty estimator to predict $\mathcal{L}_{l_2}(\boldsymbol{x}_i)$, the correlation with the ordering induced by $\mathcal{L}_{l_2}(\boldsymbol{x}_i)$ (Kendall-Tau=0.73) increases but at the same time it correlates significantly less strongly with the order based on $\mathcal{L}_{disc}(\boldsymbol{x}_i)$ (Kendall-Tau=0.56).

When we train with $\mathcal{L}_{disc}(\boldsymbol{x}_i)$, we still get a relative strong correlation with the ordering induced by $\mathcal{L}_{disc}(\boldsymbol{x}_i)$ (Kendall-Tau=0.62), but much less so with the $\mathcal{L}_{l_2}(\boldsymbol{x}_i)$ order (Kendall-Tau=0.34).

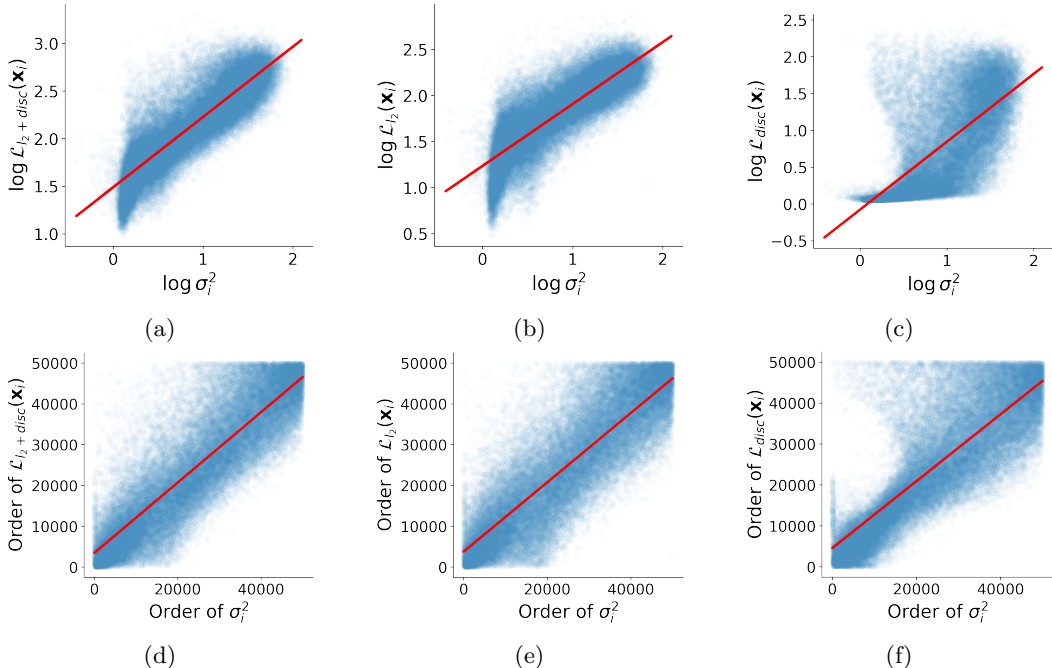

Figure 14: a, b, c) We show that the predicted $\log \sigma_i^2$, the output of $\psi_\vartheta$ on gallery images, and $\log \mathcal{L}_{l_2+disc}(\boldsymbol{x}_i)$, $\log \mathcal{L}_{l_2}(\boldsymbol{x}_i)$, and $\log \mathcal{L}_{disc}(\boldsymbol{x}_i)$, respectively, are are well correlated. c) We sort $\boldsymbol{x}_i$ once by $\sigma_i^2$ and once by $\mathcal{L}_{l_2+disc}(\boldsymbol{x}_i)$, $\mathcal{L}_{l_2}(\boldsymbol{x}_i)$, and $\mathcal{L}_{disc}(\boldsymbol{x}_i)$, respectively, to get two sets of orderings. Here, for each $\boldsymbol{x}_i$ we plot its order in the first ordering vs its order in the second ordering, and observe great correlation (Kendall-Tau correlation=0.67 (0.65, and 0.63 respectively)).

## F  Backfilling Order Analysis - VGGFace2-Gender

We now analyse which types of classes are backfilled first using our FastFill method on the VGGFace2-Gender dataset. We group the classes into a minority (female) and majority group (male). As seen in Figure 15 FastFill prioritizes the minority class for early backfilling which explains the behaviour seen in Section 5.3 in the main paper.

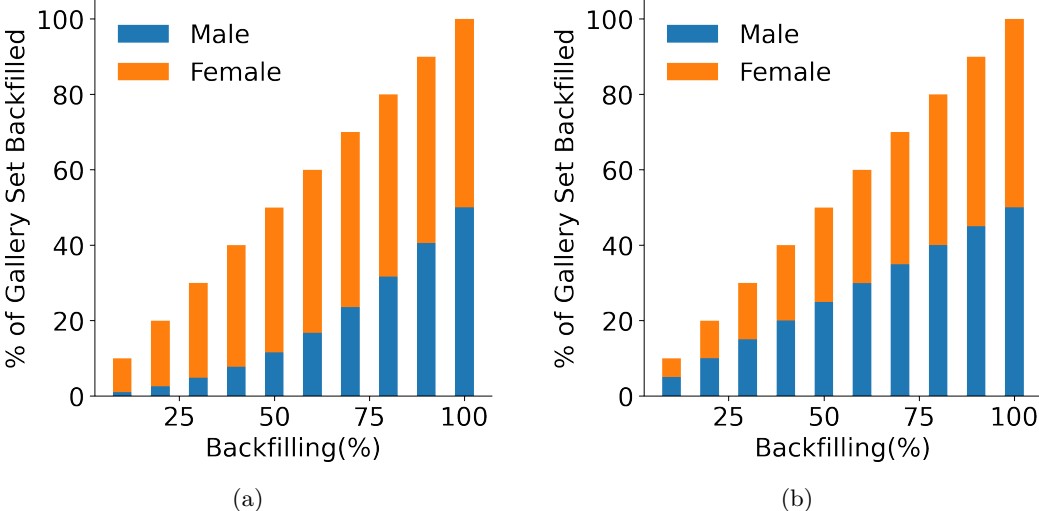

(a)                                          (b)

Figure 15: We compare the backfilling order of the minority (female) and majority (male) classes using (a) FastFill, and (b) FCT-Random. FastFill prioritizes the minority group and backfills the female gallery images first. As a results, FastFill achieves the new model accuracy gap after only $\sim 25\%$ of backfilling.

