# OpenReview forum: "FastFill: Efficient Compatible Model Update"
_ICLR.cc/2023/Conference — ICLR 2023 poster_

### Official Review · Reviewer_gEJP · 2022-10-23

**Confidence:** 5
**Correctness:** 3
**Technical Novelty And Significance:** 2
**Empirical Novelty And Significance:** 2
**Recommendation:** 3

**Clarity, Quality, Novelty And Reproducibility:**

- The authors provided the results on three standard benchmark datasets. The overall writing is clear, but not well structured. There are some missing explanations in method and experiment sections. More detailed analysis to validate the true effectiveness of proposed method is needed.
- There are neither limitation nor ethics statement section.
- This paper focuses on efficient backfilling strategy for online model upgrade, which has not been actively explored yet. However, [1] also present several simple but effective backfilling strategies but the comparisons are missing.
- I cannot evaluate on the reproducibility as there are no code implementations.

**Strength And Weaknesses:**

### Strength
- They proposed a compatible learning method for partial backfilling scenario.
- The experimental results present meaningful performance improvement on three datasets.
- The writing is easy to read.

### Weaknesses
- Novelty is weak and comprehensive ablative results are needed. [1] also presents several simple uncertainty-based backfilling strategies based on the classification score, e.g., confidence uncertainty or entropy-based uncertainty. Could you report and compare the results using these strategies on top of baseline and your model? ex) (L_l2+disc)+ confidence uncertainty or (L_l2+disc+uncert) + confidence uncertainty.
- More detailed analysis is needed. What classes of samples are first backfilled when performing the backfill strategy for each dataset? Is the class distribution uniform? or biased to some classes?
- Robustness. How is the results when the model architecture is changed, e.g., ResNet-18 to ResNet-50?
- Model bias section is weak. It presents the results on a single dataset with empirical observation only. There are some biased datasets, like as ImageNet-C, and more thorough analysis is needed to support the argument in that section.
- There are some strange experimental results but no explanations.
    - In Places-365, why new model gives worse result than old model in mAP? Even the results of mAP and CMC are conflicted.
    - In ImageNet and Places-365, why there are negative flips? Is there any ways to do early stopping?
- Why there is no BCT results on VGGFace2 and Places-365?

[1] Hot-Refresh Model Upgrades with Regression-Alleviating Compatible Training in Image Retrieval,  Zhang et al, ICLR 2022


**Summary Of The Paper:**

Image retrieval model can be updated when there is more training data or improved architecture, but it is not straightforward since old and new feature spaces are not compatible, leading to poor retrieval results between new query and old gallery embeddings. A naive solution is to replace the old features by new features from the new model, referred to as backfilling, but this may be computationally expensive and a blocker for model update. Although several compatible learning approaches have achieved compatible feature with the old model, it degrades the new model performance. To alleviate this issue, they consider two practical scenarios; 1) backfill the entire gallery set in random order and aim to maximize the average performance, and 2) backfill a partial gallery set and maximize the performance after partial backfilling. To this end, they train transformation models with uncertainty estimation using bayesian neural network and decide backfilling order based on the uncertainty of alignment loss.

**Summary Of The Review:**

This paper address the partial backfilling scenario to alleviate the limitation of existing works. The authors employ bayesian neural network to estimate the uncertainty of alignment error for efficient backfilling strategy. Although it improves the compatible performance, novelty is not strong and analysis and ablative study are insufficient, which makes the results not fully reliable.

===== POST-REBUTTAL =====

My major concern is about the true effectiveness of proposed method, which mainly consists of FCT and Bayesian-based backfilling update where FCT is borrowed from the existing work. Thus, to convince the readers to use Bayesian uncertainty for backfilling process, I still think the performance gain of FastFill should be analyzed with previous work (confidence-based uncertainty from RACT [1]) under the fair setting (FCT as backbone algorithm). If not, the readers would simply use FCT+confidence rather than FastFill, which is easier to use. However, the performance gain is marginal in the standard setting (0.6%p on ImageNet-1K and 0.2%p on Places-365 in top@1), which makes the overall novelty limited. There are more gains in disjoint setting (2.7%p in top@1 and 0.5%p in mAP on ImageNet-[500:]), but the improvements are still moderate and need to re-write the overall paper with more thorough analysis to argue it as a main claim as there are insufficient experiments. The experimental results are not sufficient to verify the need to employ additional uncertainty head and training objective and the responses do not resolve my main concerns, thus I keep my original rating.

[1] Hot-Refresh Model Upgrades with Regression-Alleviating Compatible Training in Image Retrieval,  Zhang et al, ICLR 2022

---

> ### Author Response · Authors · 2022-11-12
> **We would like to thank the reviewer for the valuable feedback and would like to address the points raised below (1/2)**
>
> We would like to thank the reviewer for the valuable feedback and would like to address the points raised below:
>
> **“Novelty is weak and comprehensive ablative results are needed. [1] also presents several simple uncertainty-based backfilling strategies based on the classification score, e.g., confidence uncertainty or entropy-based uncertainty. Could you report and compare the results using these strategies on top of baseline and your model? ex) (L_l2+disc)+ confidence uncertainty or (L_l2+disc+uncert) + confidence uncertainty.”**
>
> We proposed a novel training objective for feature alignment, an uncertainty estimation method, and a policy to order samples for partial backfilling, and we show each component is important for state-of-the-art results obtained in this work (as demonstrated in Fig. 5).
> The uncertainty estimation measures proposed in [1] (confidence and entropy) are intuitive and simple to use, but rely on the training time classification head. For most real-world image retrieval tasks (e.g. face recognition, or zero-shot retrieval) the test-time samples (for gallery and query sets) are coming from different classes from those of the training-time. Using training time classification head to measure uncertainty as in [1] is a major limitation in practice since the relative confidence/entropy between the samples from new classes tend to be not informative.  In contrast, our proposed uncertainty estimation method based on feature alignment is explicitly designed to be class agnostic and works well on previously unseen classes. This is one of our method’s main contributions/novelties.
>
> We added the ablation results you suggested (using confidence and entropy on top of our proposed loss for feature alignment and baseline FCT) using VGGFace2 dataset to Appendix F, that also helps illustrate this point. We compare different uncertainty-based backfilling ablations on i) our proposed method: L_l2+L_disc+uncert loss to train transformation, ii) FCT baseline: L_l2 loss to train transformation, and iii) RACT as in [1]. We observe that entropy/confidence based ablation on top of FastFill and baseline FCT are quite close to random backfilling whereas our proposed uncertainty estimation significantly improves them. Moreover, FastFill variants utilize transformation function from old to new feature space that leads to significantly improved model update accuracies compared to RACT which directly enforce compatibility between new and old models (more than 5% on top-1 and 25% on mAP metrics).
> We hope you reevaluate our novelty and contributions based on the provided additional experiments requested.
>
> **“More detailed analysis is needed. What classes of samples are first backfilled when performing the backfill strategy for each dataset? Is the class distribution uniform? or biased to some classes?”**
>
> We have added a more detailed analysis in Appendix E for experiments on the VGGFace2-Gender dataset. We show that FastFill uncertainty-based backfilling prioritizes samples from the female class (that was the minority class during training). Therefore, at the early stages of backfilling more females are selected which makes the class distribution skewed.
>
> **“Robustness. How is the results when the model architecture is changed, e.g., ResNet-18 to ResNet-50?”**
>
> Our experiments on VGGFace2 cover architectural changes. For the experiments on the VGGFace2 dataset, we use ResNet-18 for the old model and ResNet-50 for the new one.
>
> **“Model bias section is weak. It presents the results on a single dataset with empirical observation only. There are some biased datasets, like as ImageNet-C, and more thorough analysis is needed to support the argument in that section.”**
>
> We updated the title of section 5.3 to “Updating biased models” to better reflect our contribution and message. We have added additional results in Appendix E demonstrating that FastFill prioritizes the minority group during backfilling when updating a biased model (i.e., in our example, female subgroup data is backfilled first). Please note that this is a noticeable result, particularly given that in this experiment gallery/query classes are new (not seen during the training). FastFill reaches the bias gap of the new model after only ~25% backfilling.

---

> > ### Author Response · Authors · 2022-11-12
> > **We would like to thank the reviewer for the valuable feedback and would like to address the points raised below (2/2)**
> >
> > **“There are some strange experimental results but no explanations. In Places-365, why new model gives worse result than old model in mAP? Even the results of mAP and CMC are conflicted.”**
> >
> > Please note that in our experiment, similar to [2], our 0% backfilling corresponds to using transformed old features for gallery which is already significantly better than using old model features: old features mAP 11.6%, transformed old features mAP 18.1%,  and new features mAP 17.0%. We would like to clarify that both transformed old features (corresponding to 0% backfilling) and new features (corresponding to 100% backfilling) are significantly better than old features. New features mAP is slightly worse than transformed features mAP, which is a phenomenon that has also been observed in baseline [2]. We would like to note that this gap (1.1%) is small. In the revised paper, we include the old performances to the figures (see Fig 3).
> > mAP and CMC top-1 or top-5 regularly show slightly different results for many datasets, as mAP computes the mean of average_precision over all query images, where CMC only considers the nearest $k$ neighbours (also observed in [2]). As the difference between  new and transformed old features is small for Places-365 this makes it more likely that for mAP and CMC-top1 we get seemingly conflicting results.
> >
> > **“In ImageNet and Places-365, why there are negative flips? Is there any ways to do early stopping?”**
> >
> > Negative flips are common for different backfilling strategies as mentioned by [1]. In fact we show in Appendix C that even during the first half of backfilling where CMC performance improves significantly, there are negative flips. They are, however, outnumbered by positive flips which leads to the improvement in performance.
> > Knowing how much backfilling is sufficient (when to stop) would be very useful to save additional computation. We agree that this is an interesting new research direction which we leave to future work.
> >
> > **“Why there is no BCT results on VGGFace2 and Places-365?”**
> >
> > We have added the BCT [3] results on Places-365 (see Fig. 3 in the revised paper). FastFill outperforms BCT on Places-365. Unfortunately, the official code released by the authors does not include an implementation for the VGGFace2 dataset.
> >
> > **“There are neither limitation nor ethics statement section.”**
> >
> > We have added a limitation as well as an ethics statement to the revised version of the paper.
> >
> > **“This paper focuses on efficient backfilling strategy for online model upgrade, which has not been actively explored yet. However, [1] also present several simple but effective backfilling strategies but the comparisons are missing.”**
> >
> > We have added a comparison to the entropy based backfilling strategy proposed by [1] as an ablation study on top of FastFill, baseline FCT, and RACT in Appendix F. We observe that uncertainty-based backfilling proposed in FastFill outperforms other alternatives.
> >
> > **“I cannot evaluate on the reproducibility as there are no code implementations.”**
> >
> > We will release the code when the paper is published.
> >
> > [1] Hot-Refresh Model Upgrades with Regression-Alleviating Compatible Training in Image Retrieval, Zhang et al, ICLR 2022.
> >
> > [2] Vivek Ramanujan, Pavan Kumar Anasosalu Vasu, Ali Farhadi, Oncel Tuzel, and Hadi Pouransari. Forward compatible training for large-scale embedding retrieval systems. Proceedings of the IEEE conference on computer vision and pattern recognition, 2022.
> >
> > [3] Yantao Shen, Yuanjun Xiong, Wei Xia, and Stefano Soatto. Towards backward-compatible representation learning. In Proceedings of the IEEE/CVF Conference on Computer Vision and Pattern Recognition, pp. 6368–6377, 2020.

---

> > > ### Author Response · Authors · 2022-12-01
> > > **Looking forward to your feedback**
> > >
> > > Thank you again for your time and valuable feedback.
> > >
> > > Regarding your main concern on the novelty, we conducted all the suggested ablations and showed FastFill significantly improves over RACT [1] in all variations, and clarified differences and novelties. Particularly, we showed that FastFill addresses the general retrieval in the presence of new classes, where RACT’s uncertainty estimation policy was not informative. Please note that in all benchmarks FastFill obtains significant improvements over RACT baseline (+5% mAP on ImageNet, +16% mAP on VGGFace2).
> > >
> > > We have also added the requested analyses, and answered your other questions in the revised paper: 1) more detailed analysis on per class backfilling, 2) clarified model architecture change experiment, 3) updated description of model bias section, 4) clarified Places-365  dataset setup, 5) added BCT[3] results for Places-365 dataset, and 6) added limitations and ethics sections.
> > >
> > > We hope this revision addresses your concerns, and look forward for your response.

---

> > > > ### Comment · Reviewer_gEJP · 2022-12-01
> > > > **Response to authors**
> > > >
> > > > Thanks for your comments. Although the response addresses some of my concerns, I think the experimental results are not sufficient and thus does not resolve my main concerns. Here is the follow-up comments.
> > > >
> > > > ### Comprehensive ablative results with other uncertainty-based backfilling strategies
> > > >
> > > > There already exists some ablative study figures, like as Figure 5 or 2a on ImageNet, so the ablative results can be easily added to the figure with no extra spaces. Why you choose VGGFace2 for ablative experiments? On this dataset, the performance gain in CMC (from old to new models) are very small, so it cannot validate the effectiveness as it presents marginal gains. As well, because there are some noticeable phenomenon (like as negative flips) on ImageNet and Places-365 datasets, such ablative experiments can also verify whether such results are common or not. But these are not fully addressed.
> > > >
> > > > ### Novelty compared to RACT
> > > >
> > > > It’s hard to agree with your feedback about this issue. Even the old model is trained without classification head, the classification layer can be easily learned on top of feature extractor and its training cost would be marginal and not be different from training the uncertainty estimation head in this paper. Even the new classifier (e.g. 1000 classes on ImageNet) can be easily added to old feature extractor (trained with 500 classes). I agree that this work would have novelty if bayesian uncertainty estimation actually works well (beyond confidence/entropy based uncertainty) at the expense of training additional head, but the results are not comprehensive and thus do not fully validate it. Last but not least, this relative discussion should be in the main paper, not in the appendix.
> > > >
> > > > ### Architecture changes
> > > >
> > > > What I was curious about was how the results change when the model architectures change on the same dataset, but only one set is conducted for each dataset. Why you choose only VGGFace for architectural changes, while other datasets use the same ResNet architectures? Like the previous comments, architectural change experiments on ImageNet or Places would provide more insights.
> > > >
> > > > ### Missing Results
> > > >
> > > > The authors of FCT [1] report BCT results on VGGFace2 in Table 3. Also, this paper already reimplemented RACT on the same VGGFace2 dataset. Why did reproducing BCT on VGGFace2 and RACT on Places-365 fail?
> > > >
> > > > *Minor comments*
> > > >
> > > > Section 6 Ablation study consists of only 1 paragraph, so it may be better to include it in Section 5.
> > > >
> > > > [1] Ramanujan et al., Forward Compatible Training for Large-Scale Embedding Retrieval Systems, CVPR 2022

---

> > > > > ### Author Response · Authors · 2022-12-02
> > > > > **Additional clarifications**
> > > > >
> > > > > Thank you for your comments and feedback. Please see our responses below.
> > > > >
> > > > > **"Comprehensive ablative results with other uncertainty-based backfilling strategies":**
> > > > >
> > > > > From the three benchmarks currently in the paper (ImageNet, Places-365, and VggFace2) only the VggFace2 dataset has disjoint **test classes** than those of training (see below response to novelty). This is why we chose VggFace2 for the suggested ablation.
> > > > >
> > > > > We observe +1.6% averaged mAP improvement when using Bayesian uncertainty estimation compared to the requested ablation of **our method** using entropy. Please note that compared to the **prior work** RACT (that does not have a transformation function) with entropy based backfilling our proposed method gain is further larger: +25.9% averaged mAP.
> > > > >
> > > > >
> > > > > **"Novelty compared to RACT":**
> > > > >
> > > > > We believe there is a misunderstanding here. The main advantage of Bayesian uncertainty estimation used in FastFill is when **test time classes** (for gallery and query sets) are disjoint from **training time classes**. This is a key use-case for example in large-scale face recognition systems where test time identities are not available during training or large-scale image search systems used in search engines. Please note that we are not talking about different classes available to old vs. new models during their training. For the ImageNet dataset, the training split (and any subset of it) includes instances from the same 1000 classes of the test split.
> > > > >
> > > > > We agree with the reviewer that this is an important comparison and worth being in the main paper. We will incorporate the results currently in Appendix F into Figure 3 as a subplot in our final revision.
> > > > >
> > > > >
> > > > > **"Architecture changes":**
> > > > >
> > > > > We picked the dataset/architecture setup similar to the baseline [1] for fair comparison. They use the same old and new architectures for ImageNet and Places-365 benchmarks, but different architectures in the case of VGGFace2.
> > > > >
> > > > > Using the same dataset for old and new models, but with an architectural change as proposed above is also an interesting setup –not considered in baseline [1]. For completeness, we will report results for this particular setup on the ImageNet dataset, as suggested.
> > > > >
> > > > >
> > > > > **"Missing Results":**
> > > > >
> > > > > We have considered three prior works (FCT, RACT, and BCT) and three datasets (ImageNet, Places-365, and VGGFace2). The paper already discusses 7-out-of-9 possible cases and shows improvements with large margins. Please note that the two combinations mentioned above (BCT on VGGFace2 and RACT on Places-365) are not included in current manuscript due to 1) existence of strictly better baseline 2) absence of code/results in corresponding original works. In detail:
> > > > >
> > > > > The BCT results reported in [1] for VGGFace2 are strictly weaker compared to FCT. As reported in Table 3 in [1], the CMC top1 is 84.2% at 0% backfilling and 95.1% at 100% backfilling in BCT, while for FCT CMC top1 is 92.5% and 96.6% for 0% and 100% backfilling, respectively. For the same setup, **we have already included comparison with the strictly more accurate prior work (FCT)**, and demonstrate +1.3% mAP improvement. We will add this discussion to the paper for completeness.
> > > > >
> > > > > The RACT paper does not provide code for reproducibility. We have re-implemented their method for comparison in this work based on their description in the paper, used their suggested hyper parameters (the coefficients $\lambda$ and $\tau$), and performed grid search over several learning rate schedules to obtain the best results. Even though we could obtain significantly better results for RACT than BCT (that is a baseline for RACT) in the ImageNet experiment (+2.5% mAP), as expected, even the best setup failed to produce better results than BCT in the Places365 dataset. Therefore, we did not include this result to the figure.
> > > > >
> > > > >
> > > > > **"Minor comments":**
> > > > >
> > > > > Thanks for the suggestion. In the final version, we will make Section 6 a subsection of Section 5.
> > > > >
> > > > >
> > > > > We look forward to your feedback.

---

> > > > > > ### Comment · Reviewer_gEJP · 2022-12-03
> > > > > > **Response to authors**
> > > > > >
> > > > > > Thanks for your quick response and leave some follow-up comments.
> > > > > >
> > > > > > ### Comprehensive ablation studies
> > > > > >
> > > > > > I found that VGGFace2 has disjoint test classes, but then the results can only validate in such setting. I still cannot fully understand why the authors didn’t conduct ablative experiments on other standard benchmarks. Such experiments even do not require extra training, so the authors can easily get the results with only several times of inferences. Refer to the following paragraph for more details.
> > > > > >
> > > > > > ### Fair Comparison
> > > > > >
> > > > > > I don’t think direct comparison to RACT is fair, because RACT does not require any bottleneck for feature transformation at the beginning, unlike FCT and FastFill. Moreover, this work borrows FCT from existing work, so the authors should clarify the contribution and measure the true effectiveness of this work, on top of FCT. For example, in figure 3, there are huge initial performance gap between Fastfill(FCT) and other methods, and it seems like high initial performance comes from FCT (based on figure 3 and 5), which makes hard to evaluate. That is why I consistently ask for ablative experiments for all datasets, but still not addressed well.
> > > > > >
> > > > > > ### Novelty compared to RACT
> > > > > >
> > > > > > I agreed I had some misunderstanding about disjoint setting. I recommend adding explanations about this setting in the main paper to not confuse the readers, because it is not a common setting. However, I don’t think that confidence/entropy based uncertainty cannot be employed in this setting. If there are samples from unseen classes, then it would get uncertain classification scores, which is widely investigated in the literature of confidence calibration [1]. Because VGGFace2 results empirically validate that it gives better results than confidence-based one, I can now partially agree bayesian uncertainty works well than classification score in this setting. Without experimental results the argument is not convincing and seems to be exaggerated.
> > > > > >
> > > > > > If this paper would like to argue that the novelty is about test-time unseen classes, then this paper should more focus on it and report experimental results on more datasets in the setting. For example, the authors can split ImageNet and generate such setting.
> > > > > >
> > > > > > [1] Minderer et al., Revisiting the Calibration of Modern Neural Networks, NeurIPS 2021
> > > > > >
> > > > > > ### Missing Results
> > > > > >
> > > > > > I understand the response. I just asked because different baselines for each dataset (even without any explanation) may give readers an impression that such baselines are selected. I won’t raise further issues about this.

---

> > > > > > > ### Author Response · Authors · 2022-12-07
> > > > > > > **Response to Reviewer gEJP (1/2)**
> > > > > > >
> > > > > > > Thank you for your response and feedback. Please see our response and additional results below.
> > > > > > > ***
> > > > > > > ## "Comprehensive ablation studies"
> > > > > > >
> > > > > > > We added three new suggested ablations:
> > > > > > >
> > > > > > > 1. We performed the previously suggested **architectural change experiment** on the ImageNet dataset: **Old model is a ResNet-18, new model is a ResNet-50**, and they are both **trained on the same full ImageNet-1k training set**. Here are the average metrics over the backfilling curve  (each result is also averaged over 5 runs with different random seeds):
> > > > > > > |Method| `top-1 %`|`mAP %`|
> > > > > > > |-|:-:|:-:|
> > > > > > > |Old model|54.04|26.41|
> > > > > > > |**FastFill**|**67.77**|**45.94**|
> > > > > > > |[FCT](https://arxiv.org/abs/2112.02805)|64.91|43.43|
> > > > > > > |[RACT](https://arxiv.org/abs/2201.09724)| 64.80|42.04|
> > > > > > > |[BCT](https://arxiv.org/abs/2003.11942)|59.26|36.56|
> > > > > > >
> > > > > > > The overall trends are the same as the other setups in the original paper. A noticeable difference is that the RACT method works better in this setup, obtaining average metrics similar to FCT without requiring a transformation function. We will add the above table, full backfilling curve plots, and discussion corresponding to these experiments to the final paper.
> > > > > > >
> > > > > > > 2. We performed the suggested experiment **backfilling evaluation using other uncertainty measures for ImageNet-1k** (entropy, confidence margin, and least confidence suggested in the RACT paper) **with the same train and test time classes** (same as Section 5.2 experiment): Old and new models are ResNet-50 trained on ImageNet-500 and ImageNet-1k, respectively. Note that for evaluation, all 50,000 images in the validation set split are used for both query and gallery sets, i.e., **same classes as the training set**. Please see the results below (similar to other results, each value is averaged over 5 runs with different random seeds):
> > > > > > >
> > > > > > > |Method|Transformation|Uncertainty Est.|`top-1 %`|`mAP %`|
> > > > > > > |-|:-:|:-:|:-:|:-:|
> > > > > > > |Old model|**X**|**X**|54.04|26.41|
> > > > > > > |FastFill|✔|Bayesian|**66.49**|**44.84**|
> > > > > > > |FastFill-ablation|✔|Entropy|65.87|44.34|
> > > > > > > |FastFill-ablation|✔|Margin of Confidence|65.83|44.09|
> > > > > > > |FastFill-ablation|✔|Least Confidence|65.92|44.27|
> > > > > > > |[FCT](https://arxiv.org/abs/2112.02805)|✔|**X**| 62.80|40.47|
> > > > > > > |[RACT](https://arxiv.org/abs/2201.09724)|**X**|Entropy|60.91|39.47|
> > > > > > >
> > > > > > > In the same train and test time classes setup, FastFill obtains **+0.6% to +0.7% top-1** and **+0.5% to +0.8% mAP** improvement compared to its ablations using alternative uncertainty based backfilling policies. FastFill obtains major improvements compared to the prior works: **+3.7% top-1 and +4.4% mAP compared to FCT** baseline (which has a transformation, but no uncertainty estimation policy), and **+5.8% top-1 and +5.4% mAP compared to RACT** baseline (which does not have a transformation, but exploits entropy-based uncertainty estimation). We will add the above results to the paper.
> > > > > > >
> > > > > > > Please note that this is a setup with **the same classes in query/gallery sets as those in the training set** (as opposed to the VGGFace2 dataset case presented in our earlier response).
> > > > > > >
> > > > > > >
> > > > > > > 3. As suggested, we also show a disjoint train and test time classes setup on the ImageNet dataset: **old and new models are ResNet-50 architectures trained on ImageNet-250 and ImageNet-500, respectively**. We considered three evaluations:
> > > > > > > * **`[:500]`** Use the first 500 classes for both query and gallery. **This is the same train-test classes setup**.
> > > > > > > * **`[500:]`** Use the second 500 classes for both query and gallery. **This is the disjoint train-test classes setup**.
> > > > > > > * **`[:1000]`** Use all 1000 classes for both query and gallery. **This is a setup where half of the train-test classes overlap.**.
> > > > > > >
> > > > > > > Here are the results (averaged over 5 runs with different random seeds):
> > > > > > >
> > > > > > > |Method|Trans.|Uncert. Est.|`top-1 % [:500]`|`top-1 % [500:]`|`top-1 % [:1000]`|
> > > > > > > |-|:-:|:-:|:-:|:-:|:-:|
> > > > > > > |Old model| **X**| **X**|50.7|12.41|29.32|
> > > > > > > |FastFill|✔| Bayesian|**75.27**|**17.9**|**43.0**|
> > > > > > > |FastFill-ablation|✔| Entropy|74.26|15.22|40.78|
> > > > > > > |FastFill-ablation|✔| Margin of Confidence |74.21|15.17|40.66|
> > > > > > > |FastFill-ablation|✔| Least Confidence|74.28| 15.18|40.74|
> > > > > > > |[FCT](https://arxiv.org/abs/2112.02805)|✔| **X**|71.15|14.94|38.77|
> > > > > > > |[RACT](https://arxiv.org/abs/2201.09724)| **X**| Entropy|66.88|13.65|36.65|
> > > > > > >
> > > > > > > The above results show that Bayesian estimation brings further improved accuracy in the presence of new classes. For example, compared to its ablations, FastFill obtains **+1% top-1 improvement when evaluated on the first 500 classes `[:500]`**, but gets **+2.7% top-1 improvement when evaluated on the second 500 classes `[500:]`** that are unseen during training. Further, when evaluated on unseen classes, ablations of FastFill with logits-based uncertainty estimation gets results close to no uncertainty estimation at all as in FCT. We will add the above table, the corresponding backfilling curves, and the discussion to the paper.

---

> > > > > > > > ### Author Response · Authors · 2022-12-07
> > > > > > > > **Response to Reviewer gEJP (2/2)**
> > > > > > > >
> > > > > > > > ## "Fair Comparison"
> > > > > > > >
> > > > > > > > It is true that RACT does not require a transformation phase (a small initial cost as shown in the FCT paper).
> > > > > > > > There are trade-offs between different methods.
> > > > > > > > For example, RACT and BCT require a modified training for the new model (which is not needed in FCT and FastFill) that sometimes constrains the accuracy of the new model. For completeness, we provide comparison to all prior works while discussing their differences (and will clarify it further).
> > > > > > > > In the paper, we discussed these differences (using transformation, or a modified new model) in the text and legend of Fig. 3.
> > > > > > > > As suggested, to further clarify we will also add these differences to the ablation tables as shown above.
> > > > > > > >
> > > > > > > > Please note that the proposed method, FastFill, consists of both a new transformation training objective (pairwise L2 + a discriminative loss + Bayesian uncertainty formulation), that is different from FCT transformation training objective, and a new backfilling process using estimated uncertainties (non existent in the FCT paper). Ablation experiments demonstrate that both components are critical for improved results obtained by FastFill compared to FCT. This results in **significant improvement compared to FCT on all datasets: +4.4%, +1.3%, and +2.7% mAP improvements on ImageNet, VGGFace2, and Places365 datasets, respectively**. In addition, in the **Appendix B.1** we compare FastFill to FCT in the presence of side-information (as proposed in the FCT paper) for ImageNet setup and demonstrate clear improvement **(+3.7% mAP)**.
> > > > > > > >
> > > > > > > > ***
> > > > > > > > ## "Novelty compared to RACT"
> > > > > > > >
> > > > > > > > We will further clarify train-time vs test-time classes overlap for different experiments in the paper.
> > > > > > > >
> > > > > > > > We would like to emphasize that the novelty of the proposed work compared to RACT is not only Bayesian uncertainty estimation vs logits-based uncertainty estimation. Here is a summary of our main technical and empirical novelties compared to the RACT paper:
> > > > > > > > 1. We propose training a transformation function using pairwise and discriminative losses – non-existent in RACT (and different from FCT).
> > > > > > > > 2. In contrast to RACT, FastFill does not require modifying the training process of the new model. As discussed above this could limit the new model accuracy as we observe in some of the experiments.
> > > > > > > > 3. We model the uncertainty of the transfer function using Bayestian formulation. Ablations show that the backfilling policy based on the proposed uncertainty improves (compared to using logits-based confidence) accuracy both for same and disjoint (where gains are more emphasized) train-test time classes.
> > > > > > > > 4. On various datasets and setups FastFill obtains significant empirical gain compared to RACT.
> > > > > > > >
> > > > > > > >
> > > > > > > > ***
> > > > > > > > We thank you again for your feedback and hope the discussion and additional results provided during the rebuttal period have addressed your concerns, and appreciate it if you re-evaluate your rating of the paper.

---

> > > > > > > > > ### Comment · Reviewer_gEJP · 2022-12-09
> > > > > > > > > **Response to Authors**
> > > > > > > > >
> > > > > > > > > Thanks the authors for additional experiments. Although it’s somewhat hard to figure out the exact curve plots, I can now roughly evaluate the true effectiveness of proposed method; it provides marginal performance gain in the standard setting and more considerable margins in test-time disjoint setting, on top of FCT+confidence/entropy. I hope the full curves would be added in the paper and I think the promising direction is to more focus on test-time disjoint one.
> > > > > > > > >
> > > > > > > > > But still there are some incomplete points for verifying the robustness of results.
> > > > > > > > > 1. Why there is no ablative results on the Places-365 dataset? (It doesn't require extra training as mentioned earlier.)
> > > > > > > > > 2. Why there is no mAP results on the ImageNet-250 or ImageNet-500 datasets?
> > > > > > > > > 3. Similar overshooting (or negative flips) phenomenon occurs in the ablative models of FastFill?
> > > > > > > > >
> > > > > > > > > I am willing to raise the score based on the complete ablative results (which I have asked consistently), even without significant improvements, but it’s like repeating the same question. (I even cannot find the statement that the missing ablative results will be added to the paper.)

---

> > > > > > > > > > ### Author Response · Authors · 2022-12-10
> > > > > > > > > > **Response to Reviewer gEJP**
> > > > > > > > > >
> > > > > > > > > > In the rebuttal comments, we tried to keep the results concise and focussed to the questions raised. Moreover, we tried to provide answers without delay to have interactive discussion– even just running backfilling evaluation for different ratios of backfilling / different ablations / different random seeds / datasets takes over days. **Re: Statement: Absolutely, we will be happy to include all the detailed curves for all asked additional ablations in the final version.**
> > > > > > > > > >
> > > > > > > > > > ***
> > > > > > > > > >
> > > > > > > > > > Here are the answers for your questions:
> > > > > > > > > >
> > > > > > > > > > 1. We have provided requested ablation studies on the Places-365 dataset below. The conclusion is similar to the ImageNet ablation (**same test-train classes**) where FastFill improves its ablations using alternative uncertainty measures and significantly improves previous work FCT:
> > > > > > > > > >
> > > > > > > > > > |Method|Transformation|Uncertainty Est.|`top-1 %`|`mAP %`|
> > > > > > > > > > |-|:-:|:-:|:-:|:-:|
> > > > > > > > > > |Old model|**X**|**X**|29.57|11.39|
> > > > > > > > > > |FastFill|✔|Bayesian|**38.32**|**19.48**|
> > > > > > > > > > |FastFill-ablation|✔|Entropy|38.12|18.6|
> > > > > > > > > > |[FCT](https://arxiv.org/abs/2112.02805)|✔|**X**| 35.73|16.81|
> > > > > > > > > >
> > > > > > > > > > 2. We did not include mAP to the rebuttal comment to avoid clutter and the conclusion/trend is the same as the top-1 metric. Please see the full results for the ImageNet-250-to-500 experiment below:
> > > > > > > > > >
> > > > > > > > > > |Method|Trans.|Uncert. Est.|`top-1 % [:500]`|`top-1 % [500:]`|`top-1 % [:1000]`|`mAP % [:500]`|`mAP % [500:]`|`mAP % [:1000]`|
> > > > > > > > > > |-|:-:|:-:|:-:|:-:|:-:|:-:|:-:|:-:|
> > > > > > > > > > |Old model| **X**| **X**|50.7|12.41|29.32|29.13|2.47|14.9|
> > > > > > > > > > |FastFill|✔| Bayesian|**75.27**|**17.9**|**43.0**|**53.88**|**4.68**|**26.44**|
> > > > > > > > > > |FastFill-ablation|✔| Entropy|74.26|15.22|40.78|53.48|4.13|25.74|
> > > > > > > > > > |FastFill-ablation|✔| Margin of Confidence |74.21|15.17|40.66|53.21|4.01|25.56|
> > > > > > > > > > |FastFill-ablation|✔| Least Confidence|74.28| 15.18|40.74|53.38|4.07|25.65|
> > > > > > > > > > |[FCT](https://arxiv.org/abs/2112.02805)|✔| **X**|71.15|14.94|38.77|48.61|3.7|23.4|
> > > > > > > > > > |[RACT](https://arxiv.org/abs/2201.09724)| **X**| Entropy|66.88|13.65|36.65|41.6|3.3|20.4|
> > > > > > > > > >
> > > > > > > > > > 3. We observe very small overshooting/negative-flips towards the end of backfilling when considering other measures of uncertainty (e.g., Entropy) for ImageNet and Places experiments. Please note that prior work RACT observes negative flips in most of their results as well, but at the earlier stages of backfilling (see Fig 4, 5, and 6 in the RACT paper for example). We will add all the ablation curves in the final paper.
> > > > > > > > > >
> > > > > > > > > > ***
> > > > > > > > > >
> > > > > > > > > > We hope that the provided information answers your questions and appreciate your re-evaluation of the rating.

---

> > > > > > > > > > > ### Comment · Reviewer_gEJP · 2022-12-13
> > > > > > > > > > > **Final comment**
> > > > > > > > > > >
> > > > > > > > > > > Thanks for the response. As I mentioned before, to convince the readers to use Bayesian uncertainty for backfilling process, I still think the performance gain of FastFill should be validated on top of a combination of existing works (FCT + confidence-based uncertainty). If not, the readers would simply use FCT+confidence rather than FastFill. However, the performance gain is marginal in the standard setting (0.6%p on ImageNet-1K and 0.2%p on Places-365 in top@1), which makes the overall novelty limited. There are more gains in disjoint setting (2.7%p in top@1 and 0.5%p in mAP on ImageNet-[500:]), but the improvements are still moderate and need to re-write the overall paper with more thorough analysis to argue it as a main claim.
> > > > > > > > > > >
> > > > > > > > > > > That being said, assuming that all relative results will be added to the paper, I would raise my score to 4 because of the authors' effort for providing ablative results that enable to measure the true effectiveness.

---

> > > > > > > > > > > > ### Author Response · Authors · 2022-12-13
> > > > > > > > > > > > **Response to Reviewer gEJP**
> > > > > > > > > > > >
> > > > > > > > > > > > Thank you for your reply. Please see our responses below:
> > > > > > > > > > > >
> > > > > > > > > > > > - *"validated on top of a combination of existing works (FCT + confidence-based uncertainty)."*
> > > > > > > > > > > >
> > > > > > > > > > > > **We have provided 5 experiments for this comparison** (one already added to the paper) that has been asked during the rebuttal period (on VGGFace2, Places365, and 3 setups on ImageNet), and observed better results for FastFill compared to **its ablation using entropy** in all cases. Please note that there is no previous work with transformation + uncertainty estimation.
> > > > > > > > > > > >
> > > > > > > > > > > > - *"readers would simply use FCT+confidence rather than FastFill."*
> > > > > > > > > > > >
> > > > > > > > > > > > "FCT+confidence" is a non-existent work in the literature. The proposed method has significant improvement to this combination (which is its ablation) without introducing any additional overhead during training or inference.
> > > > > > > > > > > >
> > > > > > > > > > > > - *"the performance gain is marginal in the standard setting"*.
> > > > > > > > > > > >
> > > > > > > > > > > > Please note that the setup where test time classes are new *is the most common setup* in large-scale retrieval/recognition/search systems. Same train-test classes setup is considered standard for the classification task. Further, the proposed method, even in same train-test classes setup has clear improvement to **its ablation** and a very large margin to all prior works.
> > > > > > > > > > > >
> > > > > > > > > > > > - *"the improvements are still moderate"*
> > > > > > > > > > > >
> > > > > > > > > > > > The improvements compared to all existing works and for all setups are significant with a large margin. E.g, compared to FCT, the previous state of the art, FastFill gets  $+4.4\%$ (ImageNet), $+2.7\%$ (Places-365), and $+1.3\%$ (VGGFace2) mAP improvements.
> > > > > > > > > > > >
> > > > > > > > > > > > - *"need to re-write the overall paper with more thorough analysis to argue it as a main claim."*
> > > > > > > > > > > >
> > > > > > > > > > > > Please note that the proposed method has several components and novelties (new transformation training, new uncertainty estimation, and new backfilling policy). We have demonstrated the importance of all these technical novelties in the ablations already in the paper. The proposed method results in empirical novelty with significant margin to prior works as reported in the paper.
> > > > > > > > > > > > In addition, part of the new requested ablations are already in the paper. We will add the additional results to the paper (as detailed in our previous response) with minor changes to the current paper (sub figure in Fig. 3 and one additional curve to Fig 5).

---

> > > > > > > > > > > > > ### Comment · Reviewer_gEJP · 2022-12-13
> > > > > > > > > > > > > **Follow-up comment**
> > > > > > > > > > > > >
> > > > > > > > > > > > > Thanks for your response and here is the follow-up comment.
> > > > > > > > > > > > >
> > > > > > > > > > > > > > The proposed method has significant improvement to this combination (which is its ablation) without introducing any additional overhead during training or inference.
> > > > > > > > > > > > >
> > > > > > > > > > > > > I don't think 0.6%p and 0.2%p of gains are statistically significant. The paper mainly argues that Bayesian uncertainty estimation is important for the policy of backfilling update. However, if a simple classification score can achieve competitive results to Bayesian uncertainty, then it does not support the main argument of this paper. This is my major concern and the results in the responses do not resolve my concern. I agreed FCT+confidence is non-existent, but FastFill mostly borrows backbone algorithm from FCT and thus the actual comparison is only with confidence-based one from RACT. More importantly, because FCT+confidence is more straightforward yet achieves competitive performance, which even does not require additional head for uncertainty estimation nor complicated training objectives, it may be more solid to argue FCT+confidence as the main algorithm, rather than Bayesian estimation.
> > > > > > > > > > > > >
> > > > > > > > > > > > > > the setup where test time classes are new is the most common setup in large-scale retrieval/recognition/search systems.
> > > > > > > > > > > > >
> > > > > > > > > > > > > > even in same train-test classes setup has clear improvement to its ablation
> > > > > > > > > > > > >
> > > > > > > > > > > > > It's also hard to agree with this statement that test-time new is the most common, because all previous papers (including BCT, FCT, RACT) do not even state this setting in their papers. (Let me know if I missed some relevant results.) It's novel to introduce such setting in this task, but it does not mean that existing standard setting is not the most common. Again, it's hard to conclude that <1% of gain is a clear improvement.
> > > > > > > > > > > > >
> > > > > > > > > > > > > > need to re-write the overall paper with more thorough analysis to argue it as a main claim
> > > > > > > > > > > > >
> > > > > > > > > > > > > This statement is about test-time unseen class setting. This paper does not provide explanations about the new setting (without rebuttal I even could not figure out this setting.), and needs to conduct all analysis in the main paper on all datasets with the new setting if the authors would like to argue this setting as a main contribution.

---

> > > > > > > > > > > > > > ### Author Response · Authors · 2022-12-13
> > > > > > > > > > > > > > **Response to Reviewer gEJP**
> > > > > > > > > > > > > >
> > > > > > > > > > > > > > Thank you for your follow-up comment. Please see our note below:
> > > > > > > > > > > > > >
> > > > > > > > > > > > > > - Smaller accuracy improvements are in **comparison to an unpublished ablation of our method only for cases when train and test classes are the same**. As you mentioned, [RACT](https://arxiv.org/abs/2201.09724) is the only prior work with uncertainty estimation for comparison, on top of which we get improvements with very large margins (**up to +25.9% mAP**).
> > > > > > > > > > > > > >
> > > > > > > > > > > > > > - A setup with different train and test time classes **(VGGFace2) is in our original submission** and has **been reported in previous model compatibility work** [FCT](https://arxiv.org/abs/2112.02805). Evaluating on unseen identities is the standard protocol in [face recognition benchmarks](https://paperswithcode.com/task/face-recognition#benchmarks).
> > > > > > > > > > > > > >
> > > > > > > > > > > > > > - The proposed process has **no training time overhead compared to FCT**, and comes with **simple and efficient implementation** as highlighted by [Reviewer bcVj](https://openreview.net/forum?id=rnRiiHw8Vy&noteId=0bOzVeuw32) and [Reviewer zbYY](https://openreview.net/forum?id=rnRiiHw8Vy&noteId=6dvU9TuGDq). At inference time, with FastFill the uncertainty signal $\sigma$ is obtained by a single $d$-dimensional layer (where $d$ is the penultimate layer size, e.g., = **2048 parameters**), whereas to get entropy we require keeping a $d \times k$ dimensional tensor to get probabilities on all $k$ training time classes (in case of VGGFace2 it means a tensor of size $2048 \times 8631 =$  **17.7M parameters** -- same size order as full backbone and growing for real-world systems with more classes/identities). Aside from **lower accuracy of using entropy**, it comes with **more memory footprint and runtime computation**.

---

### Official Review · Reviewer_zbYY · 2022-10-24

**Confidence:** 5
**Correctness:** 3
**Technical Novelty And Significance:** 2
**Empirical Novelty And Significance:** 3
**Recommendation:** 6

**Clarity, Quality, Novelty And Reproducibility:**

Although the proposed training loss is somehow lacking novelty (just a sum of l2 loss and cross entropy loss), it is good for the real-product system to reproduce the method. The paper rises a new problem in compatible learning: backfilling order problem. It is a new direction to look into, and the paper also proposes a practical method to follow.


**Strength And Weaknesses:**

1. The proposed training loss is easy to implement.
2. Estimating fitting errors of gallery images and re-ordering the backfilling process is novel to compatible learning and easy for practicing.
3. The paper writing is easy to read.

**Summary Of The Paper:**

This paper proposes a training loss function with a combination of  l2 loss and cross-entropy-based loss for old/new feature transformation module training. To further improve accuracy during replacing old features with new transformed ones, the paper utilizes Bayesian method to estimate the fitting error on the gallery set and backfill the features in error descending order.

**Summary Of The Review:**

Overall, I think this paper should be accepted after addressing following concerns,

(1) some details should be explained more In the experiments part, in Table 1, 2, &3, which features are we using here? Old feature? transformed old feature with $h_{\theta}$? or new features?
(2)  In Fig 5, and Fig 3. the fig what are the initial points of curves stand for? is that the accuracy of using $h_{\theta}$ to transform old feature? If so, the numbers should be incorrect.
(3) Can we test the method on some image retrieval datasets like Google Landmark? I understand that adding experiments is not required and not recommended in rebuttal, but it will be better if we have some results on image retrieval dataset.

---

> ### Author Response · Authors · 2022-11-12
> **We would like to thank the reviewer for the positive comments and would like to address the points raised below.**
>
> We would like to thank the reviewer for the positive comments and would like to address the points raised below:
>
> **“(1) some details should be explained more In the experiments part, in Table 1, 2, 3, which features are we using here? Old feature? transformed old feature with hθ? or new features?”**
>
> The results reported in Tables 1, 2, and 3 are the average performances during the backfilling process as defined in equation (6). We always use the new model features for the query set. For the gallery set, we start off using just the transformed old features (at 0% backfilling) and then as we incrementally backfill the images, we replace them with the new features. At 100% the gallery set consists of only new model features. We have added this explanation to the Figure caption.
>
> **“(2) In Fig 5, and Fig 3. the fig what are the initial points of curves stand for? is that the accuracy of using hθ  to transform old feature? If so, the numbers should be incorrect”**
>
> The initial points of the curves represent the start of the backfilling process, where the query set consists of new model features and the gallery set of transformed old features computed by the transformation $h_{\theta}$; (in the case of the BCT and RACT baselines we just used the old features without a transformation since they do not provide a mechanism to transform old features). As we go along the curve, an increasing portion of the gallery set has been backfilled, that is the transformed old features have been replaced with the corresponding new features. In the revised paper, we add this information to Fig. 3.
>
> **“(3) Can we test the method on some image retrieval datasets like Google Landmark? I understand that adding experiments is not required and not recommended in rebuttal, but it will be better if we have some results on image retrieval dataset.”**
>
> We will try to add results on Google Landmark in the final version of the paper. Unfortunately, due to the size of the dataset (V2 has 5 million images), the training and evaluation takes longer than the rebuttal time frame.
> We also would to note that Places and VGGFace2 are also commonly used to benchmark retrieval (as in [2] and [3] baselines).
>
> [2] Vivek Ramanujan, Pavan Kumar Anasosalu Vasu, Ali Farhadi, Oncel Tuzel, and Hadi Pouransari. Forward compatible training for large-scale embedding retrieval systems. Proceedings of the IEEE conference on computer vision and pattern recognition, 2022.
>
> [3] Yantao Shen, Yuanjun Xiong, Wei Xia, and Stefano Soatto. Towards backward-compatible representation learning. In Proceedings of the IEEE/CVF Conference on Computer Vision and Pattern Recognition, pp. 6368–6377, 2020.

---

### Official Review · Reviewer_frJ2 · 2022-10-24

**Confidence:** 4
**Correctness:** 3
**Technical Novelty And Significance:** 3
**Empirical Novelty And Significance:** Not applicable
**Recommendation:** 6

**Clarity, Quality, Novelty And Reproducibility:**

The proposed solution is new. The writing is clear. The authors are suggested to include the performance of (C1) in Fig. 2/3/5 as discussed in the weaknesses section.

**Strength And Weaknesses:**

Strength

[S1] Technology: the proposed method is technically sound;

[S2] Performance: FastFill demonstrates better performance for online partial backfilling on imagenet1k, vggface2, and places365

[S3] Presentation: the writing is clear


Weaknesses

The paper introduces a new evaluation setting: online partial backfilling. The goal is to deploy the features of the gallery set on-the-fly. FastFill only shows benefits in this setting. However, I’m not sure if online partial backfilling is a feasible setting or of practical interest.

i) If I understood correctly, Figure 1 (right) presents several configurations for image retrieval: (C1) old query features + old gallery features (\phi_{old}(query) + \phi_{old}(gallery)); (C2) new query features + old gallery features (\phi_{new}(query) + \phi_{old}(gallery)); (C3) new query features + new gallery features (\phi_{new}(query) + \phi_{new}(gallery)). Here FastFill is in between (C2) and (C3). Figure 1 (right) seems to indicate that (C2) has a better performance than (C1): 61.8% vs 46.6%. I’m not sure if it is always the case as there may be a large shift in the feature spaces of \phi_{old} and \phi_{new}. I would guess the performance of (C1) is in between (C2) and (C3), e.g. a horizontal line between 61.8 and 68.2. In other words, at the early stage of the online backfilling (i.e. (C2)), the updated system may have lower performance than the old one. Therefore, it may not even be deployed. Also, I believe it is necessary to include the performance of (C1) in other diagrams, e.g. Fig.2/3/5.

ii) it is also hard to determine if the FastFill model or even (C3) actually has a better performance than (C1) without updating all the gallery features. The unseen query and gallery sets are the “golden” test set that the new model never sees. While we may be able to estimate the performance of the FastFill model or (C3) on this “golden” set in the middle of the backfilling, the estimation may not be conclusive or deserves the risk. As can be seen from Fig. 3 and 5, occasionally there may be performance drops at the last stage of the backfilling (>90%).

At this moment, I’m not sure the online partial backfilling setting is convincing. This is my main concern.

***
After rebuttal:

My concern about the configurations (i.e. Weakness i) has been addressed. Regarding the online backfilling setting (i.e. Weakness ii), I agree that it is a more practical problem that may not be expected to be resolved in one research paper. I thereby raise the score to 6.

**Summary Of The Paper:**

The paper presents an online backfilling method, FastFill, for backward-compatible retrieval systems. FastFill introduces a new feature alignment loss, L_{disc}, that maps the old feature descriptor to the corresponding cluster in the new feature space. For faster deployment, FastFill proposes to backfill the gallery images on-the-fly by a learned order that indicates the “hardness” of the samples. Compared to previous approaches, FastFill demonstrates better performance for online partial backfilling on three retrieval benchmarks: imagenet1k, vggface2, and places365.

**Summary Of The Review:**

The paper proposes FastFill, an online backfilling method for backward-compatible retrieval systems. FastFill is technically sound and demonstrates better performance for online partial backfilling on several benchmarks. However, at this moment, I still have concerns about the feasibility of the online partial backfilling setting.

---

> ### Author Response · Authors · 2022-11-12
> **We would like to thank the reviewer for the positive comments and would like to address the points raised below:**
>
> We would like to thank the reviewer for the positive comments and would like to address the points raised below:
>
> **“i) ... (C2) new query features + old gallery features (\phi_{new}(query) + \phi_{old}(gallery)) … Figure 1 (right) seems to indicate that (C2) has a better performance than (C1): 61.8% vs 46.6%. I’m not sure if it is always the case as there may be a large shift in the feature spaces of \phi_{old} and \phi_{new}. I would guess the performance of (C1) is in between (C2) and (C3), e.g. a horizontal line between 61.8 and 68.2. In other words, at the early stage of the online backfilling (i.e. (C2)), the updated system may have lower performance than the old one. Therefore, it may not even be deployed.”**
>
>
> A situation where C2 is worse than C1 is possible, and has been observed by previous works that try to learn a new model that is directly compatible with the old model (for example see RACT [1] results in Fig. 3 for the VGGFace2 dataset).
>
> We use a transformation function to align old features with the new features, therefore our C2 accuracy is much higher than C1 accuracy in all our experiments (see updated Figure 3). For instance, for the CMC-top1 retrieval set-up we get the following improvements on different datasets: on ImageNet: 61.8 (C2) vs 46.6% (C1); Places365: 37.1(C2) vs 29.6%(C1); VGGFace2: 94.4C2) vs  88.4%(C1). In practice, we thus want to deploy the new features for the query images and the transformed old features for the gallery as soon as they are available as they lead to improved results compared to using old features for both query and gallery sets.
> Moreover, note that for any percentage p (from 0 to 100) of partial backfiling, in all our results (all metrics and datasets) using FastFill accuracy of the system is significantly better than old model (C1).  We measured these accuracies on a completely disjoint offline test-set as we explain below.
>
>
> **“​​Also, I believe it is necessary to include the performance of (C1) in other diagrams, e.g. Fig.2/3/5.”**
>
> We have added the C1 ($\phi_{old}$ features for both query and gallery) performance to the figures in the revised version of the paper, as suggested, to illustrate the above point. Note that Figures 2 and 5 are ablation studies (same as ImageNet experiment in Fig. 3).
>
>
> **“ii) it is also hard to determine if the FastFill model or even (C3) actually has a better performance than (C1) without updating all the gallery features. The unseen query and gallery sets are the “golden” test set that the new model never sees. While we may be able to estimate the performance of the FastFill model or (C3) on this “golden” set in the middle of the backfilling, the estimation may not be conclusive or deserves the risk. As can be seen from Fig. 3 and 5, occasionally there may be performance drops at the last stage of the backfilling (>90%).
> At this moment, I’m not sure the online partial backfilling setting is convincing. This is my main concern.”**
>
> In all our experiments, we benchmark retrieval and back-filling accuracies on a disjoint offline test-set (from the training sets) which provides us an unbiased estimator of the real-world deployment. We use a standard setup as in baselines [1], [2], and [3]. We agree that, if the C2 or C3 accuracies are not better than C1 then the system would not be deployed, similar to any other machine learning system that is validated on an offline test benchmark. We would appreciate your feedback if more clarification needs to be added to the paper.
>
> [1] Hot-Refresh Model Upgrades with Regression-Alleviating Compatible Training in Image Retrieval, Zhang et al, ICLR 2022.
>
> [2] Vivek Ramanujan, Pavan Kumar Anasosalu Vasu, Ali Farhadi, Oncel Tuzel, and Hadi Pouransari. Forward compatible training for large-scale embedding retrieval systems. Proceedings of the IEEE conference on computer vision and pattern recognition, 2022.
>
> [3] Yantao Shen, Yuanjun Xiong, Wei Xia, and Stefano Soatto. Towards backward-compatible representation learning. In Proceedings of the IEEE/CVF Conference on Computer Vision and Pattern Recognition, pp. 6368–6377, 2020.

---

> > ### Comment · Reviewer_frJ2 · 2022-11-26
> > **Thank you for the rebuttal.**
> >
> > I'd like to thank the authors for the rebuttal. My concern about the configurations (i.e. Weakness i) has been addressed. Regarding the online backfilling setting (i.e. Weakness ii), I agree that it is a more practical problem that may not be expected to be resolved in one research paper. I have updated the score to 6.

---

> > > ### Author Response · Authors · 2022-12-01
> > > **Thank you!**
> > >
> > > Thank you for your positive feedback.

---

### Official Review · Reviewer_bcVj · 2022-10-28

**Confidence:** 4
**Correctness:** 3
**Technical Novelty And Significance:** 3
**Empirical Novelty And Significance:** 3
**Recommendation:** 8

**Clarity, Quality, Novelty And Reproducibility:**

The paper is well written and easy to understand, the ideas are novel for this particular problem, and the work is reproducible.

**Strength And Weaknesses:**

Pros
- The paper is well written and the motivations very clear. I like the idea of using the classifier head and a classification loss to guide the error of the alignment towards the same-class clusters and I find it novel.
- The authors justify their claims with thorough experiments and analysis; it includes a comprehensive evaluation on public datasets where the proposed framework is showed to outperform state of the art methods.

Cons
- I am really curious why partial backfilling in Places-365 results in better performance than after the backfilling is completed. I also wonder if any information in the policy sampling can be used to not only determine the order but also what features should not be updated.
- I personally think that the title "Reducing model biases" of 5.3 is misleading because Fastfill is not helping to reduce the biases of a model but simply "catching up" faster to the performance of the newer model, whatever bias/non-bias it has. It is an interesting experiment, but nothing different from the experiments in 5.2 and, in my opinion, slightly oversold. In other words, one can draw the same conclusions from this experiment as from the experiments presented in 5.2, and no newer ones.
- The name of the dataset use in the experiments in Section 6 is missing.

**Summary Of The Paper:**

This paper addresses the problem of backfilling in retrieval systems, which is the process of updating the features extracted from the images of your database once we have access to a newer and better model: if this is done offline, for very large databases it is a very expensive process and can take very long. The authors state that current methods of online backfilling (in which new computed features as used as soon as they are ready) in order to maintain a high accuracy during backfilling, they try to make features of the newer and older model compatible, and argue that this limits the newer model capability. They propose an online backfilling method that learns a transformation from the older feature space to the newer one (the newer model is trained independently), and a policy to order the samples that need to be backfilled first to keep the highest performance during this online process.

**Summary Of The Review:**

The proposed framework for online backfilling looks solid and novel enough. It is simple and easily applicable. The paper also has a thorough experimental analysis and  confirm the claims made in the paper.

---

> ### Author Response · Authors · 2022-11-12
> **We would like to thank the reviewer for the positive comments and would like to address the points raised below**
>
> We would like to thank the reviewers for the positive comments and would like to address the points raised below:
>
> **“I am really curious why partial backfilling in Places-365 results in better performance than after the backfilling is completed. I also wonder if any information in the policy sampling can be used to not only determine the order but also what features should not be updated.”**
>
> The question of why/when partial backfilling outperforms full backfilling in certain settings is an interesting one. In the case of Places-365, this behaviour has also been observed by the baseline Forward Compatible Training method (see Table 2 in [2]).
> Whilst the main focus of this work was for the partial backfilling performance to reach that of the fully backfilled one with as little backfilling as possible, we have some ideas of why we might get this “overshooting” behaviour.
> The fact that with FastFill partial backfilling outperforms full backfilling on some datasets seems to be in part because of the $L_{disc}$ element in our new training loss. As shown in the ablation study in Figure 5, when we train the transformation model with the $l_2$ and uncertainty only, we do not get this overshoot and neither do any of the baselines. We also do not get this behaviour on the VGGFace2 dataset, where the new model has a very high accuracy.
> Another way to explain the overshooting behaviour is by focussing on the idea of reverse backfilling. We start at the right end of the plot, where the entire gallery set consists of new model features and we want to replace some of them with transformed old features. Using the reverse FastFill ordering, we pick images with a low predicted $l_2$ and $l_{disc}$ loss. The latter part ensures that the transformation performs well on the gallery images we pick first which can lead to an increase in performance.
>
> **“I personally think that the title "Reducing model biases" of 5.3 is misleading because Fastfill is not helping to reduce the biases of a model but simply "catching up" faster to the performance of the newer model, whatever bias/non-bias it has. It is an interesting experiment, but nothing different from the experiments in 5.2 and, in my opinion, slightly oversold. In other words, one can draw the same conclusions from this experiment as from the experiments presented in 5.2, and no newer ones.”**
>
> Thanks for the feedback. We updated the title of section 5.3 as suggested to “Updating biased models” to better reflect the contribution and message. We have added additional results in Appendix E demonstrating that FastFill prioritizes the minority group during backfilling when updating a biased model (i.e., in our example, female subgroup data is backfilled first). Please note that this is a noticeable result, particularly given that in this experiment gallery/query classes are new (not seen during the training). FastFill reaches the bias gap of the new model after only ~25% backfilling.
>
> **“The name of the dataset use in the experiments in Section 6 is missing.”**
>
> The ablation study was done on ImageNet. We have added this to the plot caption.
>
> [2] Vivek Ramanujan, Pavan Kumar Anasosalu Vasu, Ali Farhadi, Oncel Tuzel, and Hadi Pouransari. Forward compatible training for large-scale embedding retrieval systems. Proceedings of the IEEE conference on computer vision and pattern recognition, 2022.

---

### Author Response · Authors · 2022-11-12
**A Summary of Changes to the Paper**

Thanks to all the reviewers for their valuable comments. We applied the following changes to the paper. Please find the detailed answers to your questions below.

- BCT result for the Places-365 dataset added.
- Class distribution results added for the experiment in section 5.3 (updating biased model) to new Appendix E.
- New variants of uncertainty-based partial backfilling results for our proposed method FastFill, and FCT and RACT baselines added to new Appendix F.
- Old model performance added to Fig. 3 and the legends updated.
- A “Limitations” section is added.
- An  “Ethics statement” is added.
- Minor text changes.

---

### Author Response · Authors · 2022-12-13
**Summary of additional results/clarifications provided in the second round (in response to Reviewer gEJP)**

Here is a summary of additional results/clarifications provided in the second discussion period (in response to reviewer gEJP). Consistent with the results and the ablations in the paper, in all cases, FastFill obtains significant gains compared to previous works and its ablations.
***
(Added to the paper in the first round:)
- **New results** to show [more detailed analysis on what classes are first backfilled](https://openreview.net/pdf?id=rnRiiHw8Vy) (See Appendix E).
- [Ethic Statement](https://openreview.net/pdf?id=rnRiiHw8Vy) (See Section 8).
- [Limitations section added](https://openreview.net/pdf?id=rnRiiHw8Vy) (See Section 9).
- **New results** to show [previous work BCT performance on Places-365 dataset](https://openreview.net/pdf?id=rnRiiHw8Vy) (See Fig. 3).
- **New results** for [ablation study on uncertainty estimation for VGGFace2 dataset with comparison to previous works RACT and FCT](https://openreview.net/pdf?id=rnRiiHw8Vy) (See Appendix F).
***
(Provided in the second round rebuttal comments, will be added in the final paper:)
- **New results** for [ablation study on uncertainty estimation for Places365 dataset with comparison to previous works FCT](https://openreview.net/forum?id=rnRiiHw8Vy&noteId=g5V0xrRGgNQ).
- **New results** for [ablation study on uncertainty estimation for ImageNet-1k dataset with comparison to previous works RACT and FCT](https://openreview.net/forum?id=rnRiiHw8Vy&noteId=zapqGYqoMVW).
- **New results** for [ablation study on uncertainty estimation for ImageNet-250-to-500 dataset with comparison to previous works RACT and FCT](https://openreview.net/forum?id=rnRiiHw8Vy&noteId=g5V0xrRGgNQ).
- [Discussion of previous work BCT on VGGFace2 dataset](https://openreview.net/forum?id=rnRiiHw8Vy&noteId=4H6_j82n9gg) (a strictly stronger prior work FCT is already included in the paper).
- [Discussion on previous work RACT re-implementation on Places365 dataset.](https://openreview.net/forum?id=rnRiiHw8Vy&noteId=4H6_j82n9gg)
- **New results** for [architectural change ablation with comparison to previous works BCT, RACT, and FCT](https://openreview.net/forum?id=rnRiiHw8Vy&noteId=zapqGYqoMVW): ResNet18 model trained on ImageNet-1k update to ResNet50 model trained on ImageNet-1k.
- [Clarifying misunderstanding on train/test time classes](https://openreview.net/forum?id=rnRiiHw8Vy&noteId=4H6_j82n9gg) (one of the novelties of FastFill).
- [Explicit statement that we will add all the above results to the final paper.](https://openreview.net/forum?id=rnRiiHw8Vy&noteId=g5V0xrRGgNQ)
***
References:
[BCT](https://arxiv.org/abs/2003.11942), [FCT](https://arxiv.org/abs/2112.02805), [RACT](https://arxiv.org/abs/2201.09724)

---

### Decision · Program_Chairs · 2023-01-20

**Decision:**

Accept: poster

**Justification For Why Not Higher Score:**

The work is rather specialized (about model update for image retrieval)

**Justification For Why Not Lower Score:**

The paper is well written and can benefit certain group of researchers (esp. image retrieval)

**Metareview: Summary, Strengths And Weaknesses:**

The paper proposes an online backfilling method that learns a transformation from the older feature space to the newer one (which is trained separately), and a policy to order the samples that need to be backfilled first to keep the highest performance during this online process.
The authors provided thorough experiments and analysis (with additional benchmarks during response), including a comprehensive evaluation on public datasets with good improvements shown. In particular,  the proposed method significantly improves over RACT which is a relevant method


**Note From Pc:**

if the above contains the word "oral" or "spotlight" please see: "oral" presentation means -> notable-top-5% and "spotlight" means -> notable-top-25%. As stated in our emails, we are disassociating presentation type from AC recommendations

**Summary Of Ac-Reviewer Meeting:**

NA, 3 out 4 reviewers are ok with the paper.
Reviewer gEJP has concern and authors have provided very reasonable responses with additional benchmarks, and hence improved the paper quality significantly